# Contributing factors to the oxidation-induced mutational landscape in human cells

Cameron Cordero [1,2,3,10], Kavi P. M. Mehta [4,5,10] ✉, Tyler M. Weaver [6,7,8,10], Justin A. Ling [6,7], Bret D. Freudenthal [6,7,8] ✉, David Cortez [4] ✉ & Steven A. Roberts [1,2,3,9] ✉

8-oxoguanine (8-oxoG) is a common oxidative DNA lesion that causes G > T substitutions. Determinants of local and regional differences in 8-oxoG-induced mutability across genomes are currently unknown. Here, we show DNA oxidation induces G > T substitutions and insertion/deletion (INDEL) mutations in human cells and cancers. Potassium bromate (KBrO$_3$)-induced 8-oxoGs occur with similar sequence preferences as their derived substitutions, indicating that the reactivity of specific oxidants dictates mutation sequence specificity. While 8-oxoG occurs uniformly across chromatin, 8-oxoG-induced mutations are elevated in compact genomic regions, within nucleosomes, and at inward facing guanines within strongly positioned nucleosomes. Cryo-electron microscopy structures of OGG1-nucleosome complexes indicate that these effects originate from OGG1's ability to flip outward positioned 8-oxoG lesions into the catalytic pocket while inward facing lesions are occluded by the histone octamer. Mutation spectra from human cells with DNA repair deficiencies reveals contributions of a DNA repair network limiting 8-oxoG mutagenesis, where OGG1- and MUTYH-mediated base excision repair is supplemented by the replication-associated factors Pol η and HMCES. Transcriptional asymmetry of KBrO$_3$-induced mutations in OGG1- and Pol η-deficient cells also demonstrates transcription-coupled repair can prevent 8-oxoG-induced mutation. Thus, oxidant chemistry, chromatin structures, and DNA repair processes combine to dictate the oxidative mutational landscape in human genomes.

Reactive oxygen species (ROS) react with nucleotide bases in DNA to form a variety of mutagenic lesions, including 8-oxoG adducts[1]. ROS are generated in cells by endogenous processes like lipid peroxidation and cell metabolism[2,3] or through exposure to exogenous agents such as potassium bromate (KBrO$_3$)[1,4], a former food additive, or UVA exposure[5]. During carcinogenesis, oncogene activation can also drive ROS formation through changes in metabolic oxidation[6]. Due to the prevalence of exogenous and endogenous agents that induce

[1]Department of Microbiology and Molecular Genetics, University of Vermont, Burlington, VT 05405, USA. [2]University of Vermont Cancer Center, University of Vermont, Burlington, VT 05405, USA. [3]School of Molecular Biosciences, Washington State University, Pullman, WA 99164, USA. [4]Department of Biochemistry, Vanderbilt University School of Medicine, Nashville, TN 37232, USA. [5]Department of Comparative Biosciences, School of Veterinary Medicine, University of Wisconsin, Madison, WI 53706, USA. [6]Department of Biochemistry and Molecular Biology, University of Kansas Medical Center, Kansas City, KS 66160, USA. [7]Department of Cancer Biology, University of Kansas Medical Center, Kansas City, KS 66160, USA. [8]University of Kansas Cancer Center, Kansas City, KS 66160, USA. [9]Center for Reproductive Biology, Washington State University, Pullman, WA 99164, USA. [10]These authors contributed equally: Cameron Cordero, Kavi P. M. Mehta, Tyler M. Weaver. ✉e-mail: kmehta@wisc.edu; bfreudenthal@kumc.edu; david.cortez@vanderbilt.edu; srober23@med.uvm.edu

oxidative stress, oxidative lesions are the second most common DNA lesion following abasic sites (AP-sites)[7]. Moreover, mutations caused by oxidative damage are a common feature of human cancer genomes with the Catalogue of Somatic Mutations in Cancer (COSMIC) single base substitution signature 18 (SBS18) hypothesized to arise from unrepaired 8-oxoG lesions. This hypothesis is supported by experimental evidence indicating that deficiency in 8-oxoG repair mechanisms[8] or UVA exposure[5], which generates cellular ROS, result in SBS18-like mutations. SBS18 occurs in ~50% of sequenced human tumors and contributes an average of 300 mutations per genome[9]. SBS18 consists of G to T (and complementary C to A) substitutions and are distributed non-uniformly across the human genome[9,10]. What factors dictate the sequence and topological specificity of 8-oxoG-induced mutation in the human genome are unknown.

Mutations caused by oxidative damage are primarily prevented by the activity of base excision repair (BER), which eliminates 8-oxoG in duplex DNA. BER can be initiated by OGG1, a glycosylase that recognizes 8-oxoG across from cytidine (C) and excises the adducted base leaving an AP-site that is subsequently processed by downstream enzymes in the BER pathway[1,11]. If OGG1 fails to remove an 8-oxoG adduct, the adducted guanidine can mutagenicaly Hoogsteen base pair with an adenine inserted by multiple polymerases (i.e. Pol δ, η, κ, and ζ) during DNA synthesis[12–15]. These 8-oxoG:dA mispairs are identified by a second DNA glycosylase, MUTYH, which cleaves the adenine leaving an AP-site that is further processed by the BER pathway[16]. Due to their direct role in removing 8-oxoG or Hoogsteen paired adenines, loss of OGG1 and MUTYH results in increased mutation rates and an altered mutation spectrum consisting of higher numbers of G to T (and complementary C to A) substitutions[8,17,18]. This elevated mutation rate is believed to cause MUTYH-associated polyposis syndrome[18], where individuals inheriting germline *MUTYH* mutations develop higher incidences of gastro-intestinal cancers throughout life[19–22]. Accordingly, human cells or cancer genomes with bi-allelic *OGG1* or *MUTYH* mutations display respective SBS18[8,18,23] and SBS36[24] mutation signatures, both of which are dominated by C to A substitutions. In other organisms, like yeast, BER can be supported by additional mechanisms to limit 8-oxoG mutagenesis. For example, some eukaryotic translesion synthesis polymerases, like Pol η, preferentially insert C across from 8-oxoG in template DNA[12,25,26], allowing error-free bypass of the lesion[27]. In addition, yeast also utilize mismatch repair to remove adenines mispaired with 8-oxoG lesions[27–29]. Whether similar alternative 8-oxoG repair mechanisms limit mutagenesis in human cells remains to be determined.

Here, we unravel the sequence and topological determinants of 8-oxoG formation, repair, and mutagenesis, and decipher how different DNA repair and tolerance pathways coordinate to produce the oxidation-induced mutational landscapes observed in human cancer genomes. We found that KBrO$_3$-treatment produces 8-oxoG lesions and mutations with similar, trinucleotide preferences indicating that ROS chemistry is the primary cause of mutational sequence specificity. Additionally, KBrO$_3$ produced a unique INDEL signature that was also observed in human cancers, providing evidence that ROS induces other mutation types beyond the canonical G > T substitutions associated with 8-oxoG. Beyond sequence determinants, we identified that chromatin structure is a key topological determinant of oxidation-induced mutations in human cancer genomes. KBrO$_3$-treatment resulted in mutations that were enriched in heterochromatin, nucleosome bound DNA, and bases facing the histone octamer, a similar phenomenon observed in the SBS18 mutational signature[30]. Cryo-EM structures of OGG1 bound to nucleosomes containing 8-oxoG revealed the enzyme uses a DNA sculpting and base flipping mechanism for repairing 8-oxoG in the nucleosome, providing a mechanistic basis for the elevated mutational density at bases facing the histone octamer. Finally, analysis of mutation spectra from OGG1-, MUTYH-, Pol η-, and HMCES-deficient cells determined the human

8-oxoG repair network includes OGG1 and MUTYH performing primary mutation avoidance while Pol η and HMCES function in secondary roles mediating tolerance of unrepaired 8-oxoG or 8-oxoG-derived AP sites. Subsequent analysis of Pol η-deficient cells has unveiled the presence of transcription-coupled repair of 8-oxoG on the transcribed strand of genes.

## Results

To characterize processes that modulate 8-oxoG mutagenesis and thereby dictate its distribution in human cells, we propagated independent clonal isolates of wild-type immortalized human retinal epithelial cells (hTERT-RPE-1) in the absence and presence of 250 μM KBrO$_3$ for 100 days, to mimic an exogenous 8-oxoG-producing exposure. RPE-1 cells were chosen for their diploid genome status, which facilitates mutation calling, and non-cancerous origin making these cells a closer model to normal cells in the body. KBrO$_3$ treatment resulted in only a modest increase in cellular ROS ( ~ 2-fold; Supplementary Fig. 1), This result is consistent with previous experiments showing that KBrO$_3$ induces 8-oxoG by a chemical reaction that requires glutathione and is resistant to catalase and superoxide dismutase ROS scavengers, therefore occurring distinctly from traditional ROS[4]. Surviving KBrO$_3$-treated clonal isolates were obtained following this outgrowth and genomic DNA was isolated for Illumina whole genome sequencing. Whole genome sequencing of outgrowth clones was compared to that of corresponding pre-outgrowth populations to identify mutations accumulated during propagation (Fig. 1A). A total of 19684 and 128366 mutations were identified from untreated and KBrO$_3$-treated cells, respectively, using the consensus calls of three probabilistic variant callers: VarScan2[31], SomaticSniper[32], and Strelka2[33]. Cells treated with KBrO$_3$ had a 23.5-fold increase in substitutions and 3.35-fold increase in small insertion/deletion (INDEL) mutations per sequenced genome when compared to the non-treated cells (Fig. 1B).

### Spontaneous and KBrO$_3$-induced mutation spectra in human retinal pigment epithelial cells

Based upon this large increase in mutation load, we assumed most mutations within the KBrO$_3$-treated cells were induced by 8-oxoG. We therefore produced a de novo KBrO$_3$ specific mutation signature using SigProfilerExtractor[34] set to detect 2 signatures from our dataset: one corresponding to the KBrO$_3$-induced mutations (SBS96A) and the second representing the spontaneously acquired mutations during untreated outgrowth (SBS96B) (Supplementary Fig. 2A, B). Deconvolution of these signatures into known COSMIC SBS signatures revealed that untreated RPE-1 cells contained a broad spectrum of base substitution mutations most consistent with SBS40, SBS5, and a small percentage of SBS18 (Supplementary Fig. 2C), which are all consistent with spontaneously accumulated mutations during cell culture[5]. Contrastingly, KBrO$_3$-treated cells were dominated by C > A substitutions, as expected for mutations derived from unrepaired 8-oxoG (Fig. 1C and Supplementary Fig. 2D).

SigProfilerExtractor also identified 2 signatures for INDEL mutations. Untreated cells contained primarily 1 bp T insertions and deletions in long homopolymer runs and thus appears to be a combination of COSMIC INDEL (ID) signatures ID1 and ID2 (Fig. 1D). These mutation signatures are associated with replication slippage events that would be expected to arise spontaneously through cell division[9]. KBrO$_3$-treated cells also contained the ID1- and ID2-like mutations, though we observed a significant number of 1 bp deletions of C and T nucleotides (Fig. 1D). These deletions were most common when not in homopolymer runs suggesting that they are likely induced by DNA damage and independent of polymerase slippage. While a 1 bp deletion of C bases could logically stem from error-prone replication past a KBrO$_3$-induced 8-oxoG, the presence of a similar number of 1 bp T

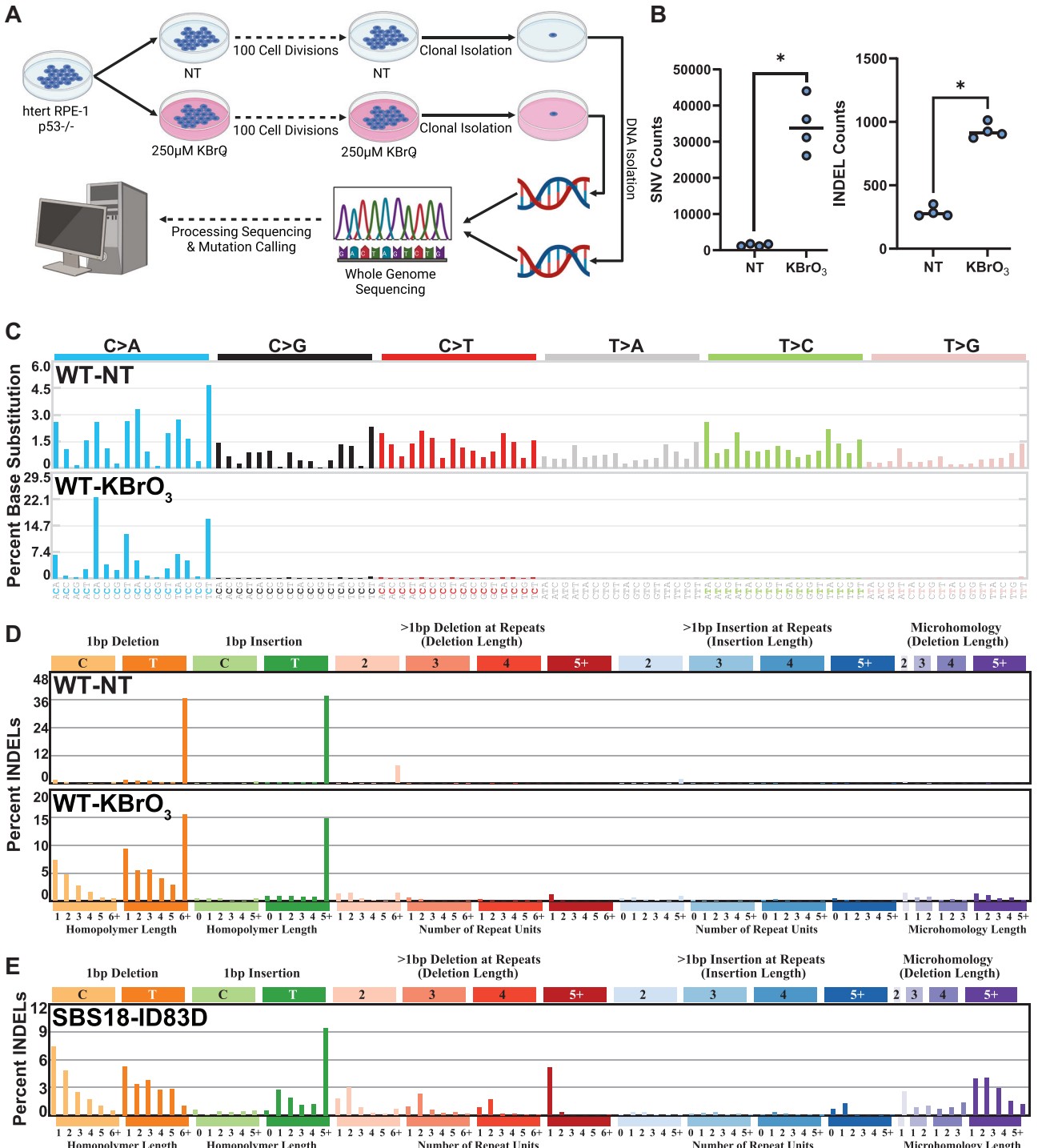

**Fig. 1 | Mutagenesis in hTERT RPE-1 p53$^{-/-}$ cells untreated and treated with 250 μM KBrO$_3$ after 100 cell divisions. A** Schematic of experimental conditions for mutation accumulation, clonal isolation, WGS, and mutation calling. Image created with BioRender.com and licensed for publication under agreement number TD26OVBCYI. **B** Number of SNVs and INDELs per genome in untreated (NT) and KBrO$_3$-treated (KBrO$_3$) cells. Circles indicate biologically independent genomes sequenced ($n = 4$). Horizontal bars are median values. (* indicates $p$-value = 0.0286 by two-sided Mann-Whitney U test comparing KBrO$_3$-treated to untreated clones) **C** SNV and **D** INDEL mutation signatures from treated and untreated genomes. **E** De novo generated INDEL signature found in human cancers containing greater than 25% of total mutations attributed to SBS18. Total mutations involved in each de novo generated INDEL signature are listed in Supplementary Fig. 4.

base deletions was surprising. KBrO$_3$ largely produces 8-oxoG lesions through a reaction with glutathione that generates an unknown oxidant[4,35], and no T based lesion has been specifically identified. The sequences flanking 1 bp T deletions displayed a random distribution of C:G and A:T base pairs, indicating that the T deletions were unlikely to be collateral mutations caused by

extended synthesis via deletion-prone TLS polymerases recruited to bypass an 8-oxoG[36] (Supplementary Fig. 3A).

We next evaluated whether a similar INDEL signature is potentially associated with endogenous ROS during cancer development. We obtained mutation calls for whole genome sequenced primary tumors from the International Cancer Genome Consortium (ICGC) and

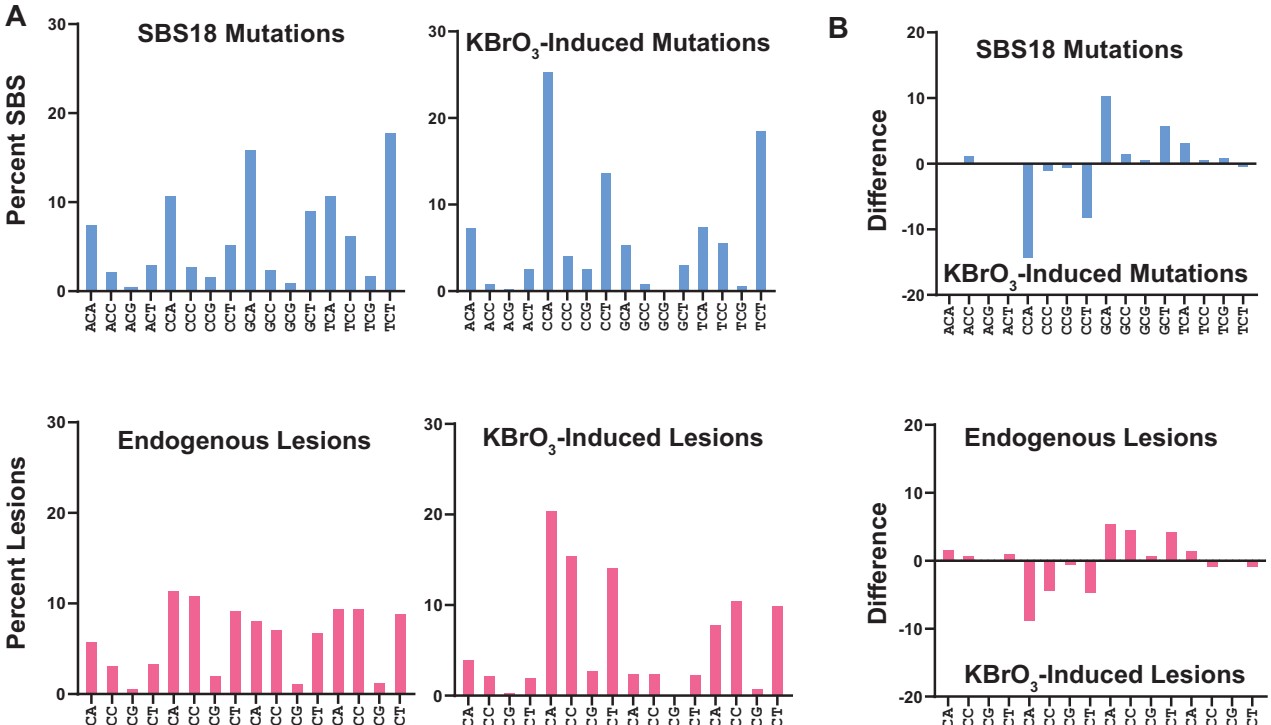

**Fig. 2 | C > A mutation spectra and 8-oxodG lesion spectra in human cells under endogenous or KBrO$_3$-induced DNA damage. A** Comparison of percentages of C > A trinucleotide mutation contexts from COSMIC SBS18 and KBrO$_3$-induced mutations gave a cosine similarity of 0.812 and comparison with endogenous 8-oxodG lesion mapping gave a cosine similarity of 0.867. Comparison of KBrO$_3$-induced mutations with KBrO$_3$-induced 8-oxodG lesions gave a cosine similarity of 0.896. **B** Differential bar graphs display discrepancies in trinucleotide contexts percentages between SBS18 and KBrO$_3$ mutations has a cosine similarity of 0.908 compared to the differential bar graphs displaying discrepancies between trinucleotide contexts from endogenous 8-oxodG lesions compared to endogenous KBrO$_3$-induced 8-oxodG lesions.

identified 68 tumors containing greater than 25% of their substitutions stemming from SBS18, meaning that ROS is a major mutagen in these samples. Subsequently, we utilized SigProfilerExtractor on these samples to produce de novo INDEL signatures. This analysis determined that the fourth-most abundant INDEL signature (constituting ~12% of INDEL mutations) had high similarity to our KBrO$_3$-induced INDEL signature (Fig. 1E and Supplementary Fig. 4; cosine similarity = 0.751). We also utilized mutationalpatterns.R[37] to re-assign mutations in ICGC to the entire catalog of COSMIC signatures with the addition of our KBrO$_3$-induced INDEL signature. Following this process, the number of mutations in our KBrO$_3$-induced INDEL signature correlated with the number of SBS18 mutations (Supplementary Fig. 3B), indicating that the two signatures are likely linked, and that endogenous ROS produces insertion/deletion mutations in addition to well characterized substitutions in tumors. Recent sequencing of normal epithelial crypts in human colon samples also observed the presence of the INDEL signature ID5 in association with SBS18 mutations[38]. The COSMIC ID5 signature closely resembles our KBrO$_3$-induced INDEL signature and correlates strongly with SBS18 in tumors (Supplementary Fig. 3C, D). These similarities strongly suggest that COSMIC ID5 is an oxidation-induced mutation signature.

### Endogenous and exogenous oxidants produce different mutation signatures

We next compared the KBrO$_3$-induced SBS signature to SBS18, which is proposed to originate from ROS producing 8-oxoG in human cancers (Fig. 2A). While both the KBrO$_3$ and SBS18 signatures were dominated by C > A substitutions, the dominant sequence contexts at which mutations occur were different, producing a cosine similarity of only 0.812. This difference was most pronounced at the sequences CCA,

CCT, GCA, and GCT. KBrO$_3$-treatment produced a greater proportion of mutations at CCA and CCT sequences and a corresponding lower proportion of mutations in GCA and GCT sequences compared to SBS18. We wondered whether this difference in sequence specificity resulted from differences in the location of 8-oxoG formation caused by endogenous ROS compared to KBrO$_3$. We therefore obtained CLAPS-seq reads, which identify the genomic positions (at single nucleotide resolution) of 8-oxoG lesions formed during culture, from HeLa cells grown in the presence and absence of KBrO$_3$[39]. Like mutations, KBrO$_3$-induced 8-oxoG occurred at a different distribution of sequence contexts compared to 8-oxoG caused by endogenous ROS in untreated HeLa cells (cosine similarity = 0.909) (Fig. 2B). KBrO$_3$-induced lesions occurred in contexts highly similar to KBrO$_3$-induced mutations, except for a higher proportion of lesions occurring in the context of CCC, suggesting that these lesions may be either more accurately bypassed or preferentially repaired prior to mutagenesis. 8-oxoG generated by endogenous ROS is also over-represented in the CCC context compared to mutations in SBS18. However, these lesions also occurred in CCT and GCC contexts more frequently than SBS18 mutations. Still, the difference of the KBrO$_3$ and endogenous 8-oxoG proportional sequence contexts displayed a striking similarity to that of the KBrO$_3$-induced mutations and SBS18, suggesting that differences in lesion formation largely account for the differences between the KBrO$_3$ and SBS18 mutation signatures.

### BER reduces 8-oxoG mutations in solvent exposed, less chromatin compacted DNA

In human cancers, mutation densities caused by a variety of DNA damages are dictated in part by chromatin structure with heterochromatic regions having higher mutation rates arising from reduced

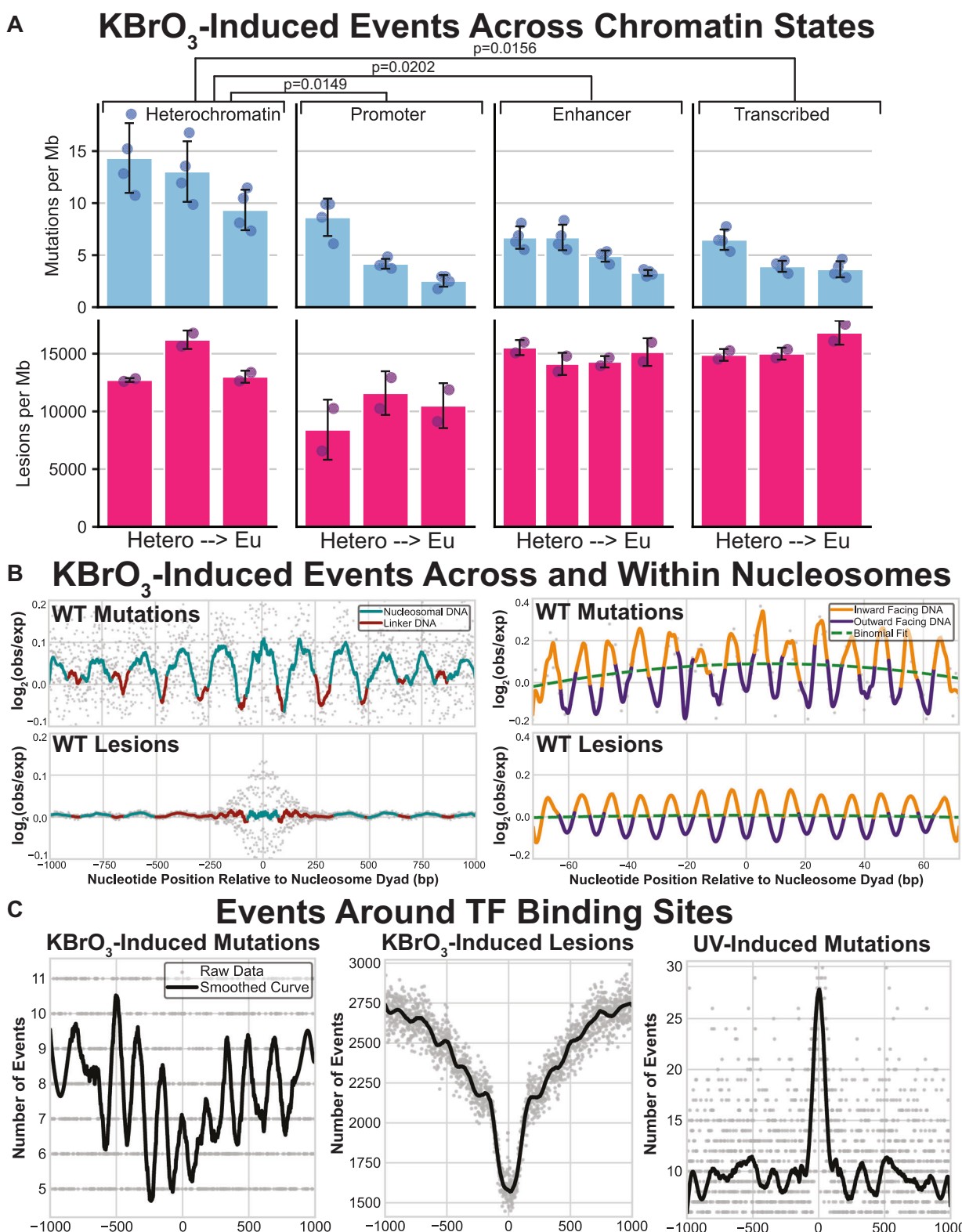

**A   KBrO₃-Induced Events Across Chromatin States**

**B   KBrO₃-Induced Events Across and Within Nucleosomes**

**C   Events Around TF Binding Sites**

DNA repair efficiency[40,41]. We sought to determine whether BER of 8-oxoG lesions was similarly impacted by chromatin resulting in a non-random distribution of mutations. To accomplish this, we profiled where KBrO₃ treatment forms 8-oxoG lesions and mutations (using HeLa cell CLAPS-seq data and WT RPE-1 cell variant calls, respectively) relative to different chromatin states derived by Hidden Markov Modeling (HMM) of eight histone modifications and

CTCF[42–44] (Fig. 3A). This modeling results in 15 chromatin states with different extents of euchromatic character. 2 of these states are associated with highly repetitive sequences and therefore were excluded from our analysis. By stratifying heterochromatin, promoters, enhancers, and transcribed regions into different states, we found that KBrO₃-induced mutations decreased in less compact regions. Interestingly, 8-oxoG lesions were evenly distributed across

**Fig. 3 | Chromatin state, nucleosome binding, and transcription factor binding's impact on 8-oxodG mutagenesis and lesion formation. A** Four binned broad regions are dictated by the ChromHMM map and within each group are sorted from left to right as being more heterochromatic to more euchromatic. The mean density of mutations and lesions are represented in blue bars (top graphs) and pink bars (bottom graphs), respectively. Error bars represent standard deviation and the circles represent biologically independent sequenced genomes ($n = 4$) or technical replicates of lesions ($n = 2$). Mutation rates in heterochromatin compared to other domains were significantly different for mutations. Precise $p$-values are indicated in the figure and were derived by Bonferroni-corrected two-sided paired t-test. **B** The left graphs represent translational periodicity of log$_2$(observed/ expected) of events between nucleosomes with mutations on top and lesion on the bottom. Nucleosome bound DNA is represented in blue and linker DNA is represented in red. The right two graphs represent the rotational periodicity of the log$_2$(observed/expected) of events within the nucleosome where DNA that is inward facing relative to the nucleosome is displayed in gold while outward facing relative to the nucleosome is displayed in purple. A binomial fit of the data is overlaid in a dashed green line. In both figures, actual data points are displayed in gray. **C** Number of events are plotted relative to the TF binding midpoint for mutations on the left, lesions in the middle and UV-induced mutations occurring in dipyrimidine contexts from sequenced melanomas[91]. Original data points are displayed in gray and a smoothed curve is overlaid in black.

all states indicating that inhibited DNA repair in heterochromatin likely underlies the higher mutation rates in these regions. As the repressive nature of heterochromatin is largely generated by tightly packed nucleosomes within these regions, we also profiled 8-oxoG mutations and lesions around strongly positioned nucleosomes within the human genome (Fig. 3B). Consistent with our prior analysis, we observed 8-oxoGs formed relatively evenly across nucleosome bound regions. However, KBrO$_3$-induced mutations oscillated with a ~192 bp periodicity peaking within histone bound DNA, while lower mutagenicity was observed in linker DNA between nucleosomes. This finding is consistent with DNA repair being inhibited by tightly bound histones that obscure access to 8-oxoG during repair. Repair inhibition also extended within individual nucleosomes as KBrO$_3$-induced mutations displayed a strong 10.3 bp oscillation (Fig. 3B). The peaks of this oscillation occurred at inward facing nucleotides closest to the histones, whereas the troughs occurred at the most solvent exposed nucleotides. This result indicates that either 8-oxoG preferentially forms at histone proximal nucleotides or that repair of 8-oxoG by BER is more efficient at solvent exposed lesions. 8-oxoG lesions displayed a similar pattern as KBrO$_3$-induced mutations, but with a significantly lesser amplitude. We therefore conclude that efficient BER at outward facing bases is the primary factor influencing 8-oxoG mutations within nucleosomes, though a subtle lesion formation preference may exist. To determine if other DNA proteins beyond histones can block the repair of 8-oxoG, we profiled KBrO$_3$ mutations and lesions at active transcription factor binding sites (Fig. 3C). We found that neither mutations nor lesions were elevated at these sites in contrast to other types of DNA damage like CPDs[45], suggesting that the inhibition of 8-oxoG repair is specific for nucleosome structure.

The mapping of 8-oxoG lesions and KBrO$_3$-induced mutagenesis indicate that 8-oxoG undergoes preferential repair at solvent-exposed positions in the nucleosome (Fig. 3B). Consistent with these findings, previous work identified that the DNA glycosylase OGG1 excises 8-oxoG from solvent-exposed positions more efficiently than histone-occluded positions in recombinant nucleosomes in vitro[46,47]. To obtain mechanistic insight into the preferential repair of solvent-exposed 8-oxoG in the nucleosome, we determined a 3.3 Å cryo-EM structure of OGG1 bound to a nucleosome containing a solvent-exposed 8-oxoG at superhelical location (SHL) − 6, referred to as OGG1-8-oxoG-nucleosome core particle (NCP) − 6 (Fig. 4A, Supplementary Figs. 5–7, and Supplementary Table 1). We utilized a catalytically dead variant of OGG1 (K249Q) that maintains the ability to specifically recognize 8-oxoG ensuring we captured an 8-oxoG substrate recognition complex[48,49]. The local resolution of the nucleosome was 3–4 Å and the local resolution of OGG1 was 5–7 Å (Supplementary Fig. 6F), which was sufficient to unequivocally dock the previously determined high-resolution X-ray crystal structure of OGG1 (PDB: 1EBM)[48] into the cryo-EM map (Supplementary Fig. 6H). Although the local resolution of OGG1 (5–7 Å) was not sufficient for determining the exact position of OGG1 side chains, the side chain conformations presented below represent those from the high-resolution X-ray crystal structure of OGG1 (PDB: 1EBM)[48].

In the OGG1-8-oxoG-NCP − 6 substrate recognition complex, OGG1 is engaged with ~5 base pairs of nucleosomal DNA at SHL − 6, which buries ~1086 Å$^2$ of surface area (Fig. 4A). The interaction of OGG1 with the nucleosomal DNA is mediated by a network of non-specific interactions with the phosphate backbone of the damaged nucleosomal DNA strand, as well as extensive contacts with the orphan cytosine and 8-oxoG (Fig. 4, B C). Interestingly, we did not observe a direct interaction between OGG1 and the histone octamer, indicating that nucleosome binding by OGG1 is primarily driven by the interactions with nucleosomal DNA. At the center of the OGG1 binding footprint lies the nucleosomal 8-oxoG, which has been evicted from the DNA helix and positioned into the OGG1 active site (Fig. 4B C). In this conformation, the extrahelical nucleosomal 8-oxoG is positioned in proximity to key OGG1 amino acid residues that are important for 8-oxoG binding specificity (G42 carbonyl), stabilization of the extrahelical 8-oxoG (C253, F319, and Q315), and 8-oxoG excision (K249Q and D268) (Fig. 4C)[48]. Cumulatively, this data shows OGG1 is in a conformation poised for 8-oxoG excision.

To position the 8-oxoG into the catalytic active site, OGG1 binding induces significant structural changes in the nucleosomal DNA during 8-oxoG recognition. These structural changes include a 1 bp register shift in the nucleosomal DNA, significant minor groove widening at SHL − 6, and nucleosomal DNA bending around SHL − 5.5 to SHL − 6.5 when compared to 8-oxoG-NCP − 6 (Fig. 4D). The OGG1-induced minor groove widening and nucleosomal DNA bending facilitate extrusion of the 8-oxoG from the nucleosomal DNA into the OGG1 active site (Fig. 4C). Ultimately, the mode of 8-oxoG recognition and the OGG1-induced structural changes in the nucleosomal DNA are similar to those seen for OGG1 bound to 8-oxoG in non-nucleosomal DNA (RMSDDNA - 1.621) (Supplementary Fig. 8A)[48], indicating a conserved 8-oxoG recognition mechanism in chromatin and non-chromatinized DNA.

To determine whether OGG1 uses the same mechanism for 8-oxoG recognition at different translational locations in the nucleosome, we determined a 3.6 Å cryo-EM structure of OGG1 K249Q bound to a nucleosome containing a solvent-exposed 8-oxoG at SHL + 4, referred to as OGG1-8-oxoG-NCP + 4 (Fig. 4E, Supplementary Figs. 9–11, and Supplementary Table 1). The local resolution of the nucleosome was 3–6 Å and the local resolution of OGG1 was 5–8 Å (Supplementary Fig. 10F). Importantly, the cryo-EM map was sufficient to dock the previously determined high-resolution X-ray crystal structure of OGG1 (PDB: 1EBM)[48] into the cryo-EM map (Supplementary Fig. 10H), and the side chain conformations were kept from the high-resolution X-ray crystal structure of OGG1 (PDB: 1EBM)[48].

The general mechanism of nucleosome binding and 8-oxoG recognition by OGG1 at SHL + 4 are similar to those observed for OGG1 bound to 8-oxoG at SHL-6 (Supplementary Fig. 8B, C). However, OGG1 binding induces modest structural rearrangements in the nucleosomal DNA during 8-oxoG recognition at SHL + 4, which includes minor groove widening of the nucleosomal DNA without significant DNA bending. The lack of OGG1-induced DNA bending is likely due to the inherently bent conformation of the nucleosomal DNA near SHL + 4 (Supplementary Fig. 8D). Despite these subtle differences, the final

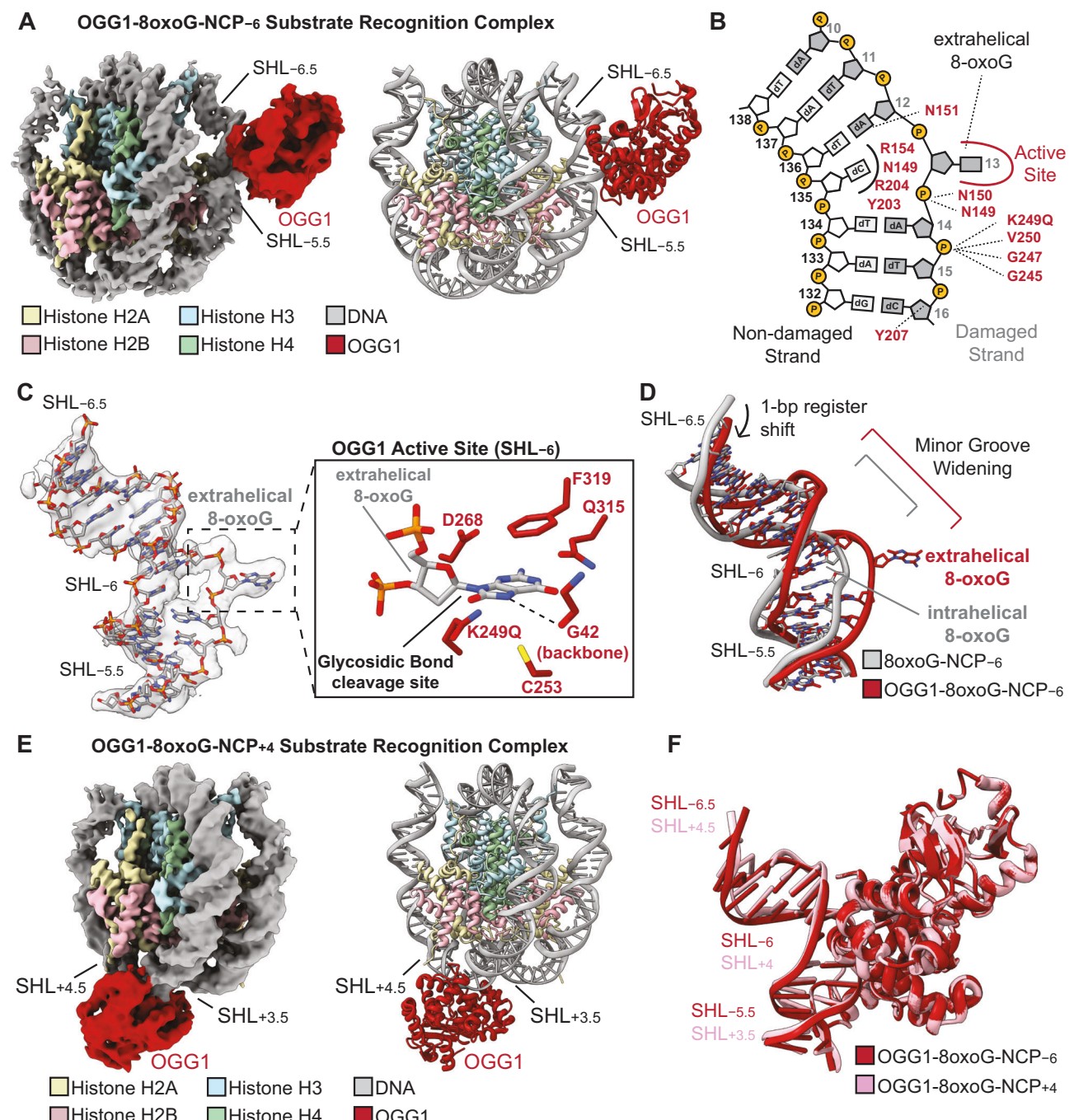

**Fig. 4 | Single particle analysis of OGG1-8-oxoG-NCP − 6. A** The 3.3 Å OGG1-8-oxoG-NCP − 6 composite cryo-EM map (left) and cartoon representation of the OGG1-8-oxoG-NCP − 6 model (right). **B** A diagram representing the interactions between OGG1 and the nucleosomal DNA in the OGG1-8-oxoG-NCP − 6 complex identified using PLIP[92]. **C** Focused view of the nucleosomal DNA at SHL − 5.5 to SHL − 6.5 showing the extrahelical 8-oxoG at SHL − 6. The segmented density for the nucleosomal DNA in the OGG1-8-oxoG-NCP − 6 composite cryo-EM map is shown in transparent grey. An inset of the OGG1 active site is shown, highlighting key amino acids important for 8-oxoG recognition and excision. **D** Structural comparison of the nucleosomal DNA (SHL − 5.5 to SHL − 6.5) in the OGG1-8-oxoG-NCP − 6 complex and 8oxoG-NCP − 6, highlighting the structural changes in the nucleosomal DNA induced by OGG1 binding. **E** The 3.6 Å OGG1-8-oxoG-NCP + 4 composite cryo-EM map (left) and cartoon representation of the OGG1-8-oxoG-NCP + 4 model (right). **F** Structural comparison of OGG1 and the nucleosomal DNA (SHL − 5.5 to SHL − 6.5) in the OGG1-8-oxoG-NCP − 6 and OGG1-8-oxoG-NCP + 4 complexes, highlighting the similarities in 8-oxoG recognition at both positions.

conformation of OGG1 and the nucleosomal DNA in the OGG1-8-oxoG-NCP + 4 and OGG1-8-oxoG-NCP − 6 structures are very similar (Fig. 4F and Supplementary Fig. 8A). This data strongly suggests that OGG1 uses the same general mechanism for 8-oxoG recognition and repair at solvent-exposed positions throughout the nucleosome. Notably, the structural changes observed during the recognition of solvent-exposed 8-oxoG by OGG1 are incompatible with binding 8-oxoG proximal to the histone octamer, as this would result in significant clashes between OGG1 and the core histone octamer (Supplementary Fig. 8E, F). Together, this data provides a strong structural rationale for the preferential repair of solvent-exposed 8-oxoG in the nucleosome in vitro and in vivo, and the elevated levels of KBrO₃-induced mutagenesis at nucleotides proximal to the histone octamer (Fig. 3B)[46,47].

## Replication and transcription-associated mechanisms limit 8-oxoG mutagenesis

In various species, 8-oxoG mutagenicity is limited by multiple, redundant DNA repair and damage tolerance pathways. This includes the activities of OGG1-initiated BER, MutY-initiated BER, mismatch repair, nucleotide excision repair (NER), and accurate TLS bypass by DNA polymerase η. We therefore compared spontaneous and KBrO₃-induced mutation spectra among WT human cell lines and those lacking OGG1[8], MUTYH[8], Pol η[50], or HMCES, a recently identified replication-associated factor that participates in bypass of ssDNA lesions[51–55] and protects cells from cytotoxicity associated with KBrO₃ exposure[51,54]. Loss of OGG1, MUTYH, and HMCES resulted in moderate ~2 to 3–fold increases in the amount of spontaneously acquired mutations per genome compared to corresponding WT lines, while Pol η-deficiency failed to increase spontaneous mutagenesis (Supplementary Fig. 12A). Changes in mutagenesis in these repair deficient cells were primarily due to increased substitutions as, spontaneous INDEL frequency was only increased in *HMCES⁻/⁻* cells, which displayed less than 2-fold increase in 1 bp T insertions and deletions (Supplementary Fig. 13). RPE-1 cells lacking HMCES maintained similar substitution spectra compared to WT cells, suggesting the increased spontaneous mutation load results from a general reduction in error-free lesion bypass (Supplementary Fig. 12B). Contrastingly, loss of either OGG1 or MUTYH produced spectra consisting primarily of SBS18-like mutations, indicating these glycosylases are the primary mechanism for preventing 8-oxoG mutagenesis and HMCES is likely involved more generally in lesion bypass (Supplementary Fig. 12B).

To directly evaluate the role of Pol η and HMCES in 8-oxoG bypass, we compared mutation spectra from RPE-1 knockout lines following prolonged KBrO₃ exposure to those in WT RPE-1 cells (evaluated in Fig. 1). Deficiency in Pol η or HMCES resulted in 1.5– and 1.3–fold increases in total KBrO₃-induced mutations, respectively, although statistical significance of increased mutation load in the Pol η knockout cells was not possible due to the lack of replicate samples sequenced (Fig. 5A). These mild increases are likely underestimates of the true augmentation of KBrO₃ mutagenesis as *HMCES⁻/⁻* cells showed a significant growth delay upon initial KBrO₃ exposure likely resulting in fewer cell divisions for these lines. Despite the lack of statistical significance for total mutations load, loss of either HMCES or Pol η significantly altered the mutation spectra for both substitutions and INDEL mutation types (Fig. 5A). The substitution spectra of Pol η- and HMCES-deficient cells were nearly identical to that of KBrO₃-treated WT cells (Fig. 5B), indicating that both enzymes likely participate in some form of error-free bypass of 8-oxoG or derived repair intermediates. Interestingly, loss of Pol η increased not only G > T substitutions predicted to be caused by 8-oxoG bypass, but also G > C, and G > A substitutions ($p < 1×10^{-5}$, by Chi-square) as well (Supplementary Fig. 14), suggesting that another polymerase may insert G or T across from 8-oxoG in Pol η's absence. Neither gene disruption changed the impact of chromatin compaction on 8-oxoG induced substitution frequency (Supplementary Figs. 15 and 16), suggesting they are primarily operating in contexts without nucleosome involvement. INDEL spectra from KBrO₃-treated Pol η- and HMCES-deficient cells, were also like that of WT cells, except for several subtle differences. *HMCES⁻/⁻* cells displayed a small increase in 1 bp T insertions at shorter homopolymer lengths, while Pol η-deficiency resulted in a general loss of 1 bp T insertions and a preference for 1 bp C or T deletions occurring in 2–3 bp homopolymer repeats (Fig. 5C). Ultimately, these differences in mutation spectra, particularly for INDELs, further supports roles for both Pol η and HMCES in oxidative lesion bypass.

To determine whether mismatch repair (MMR) limits 8-oxoG mutagenesis in human cells, we evaluated replication strand asymmetry of KBrO₃-induced mutations and 8-oxoG-induced G to T substitutions in untreated *OGG1⁻/⁻* and *MUTYH⁻/⁻* cells using methods similar to AsymTools2 software[56] that determines leading and lagging strand association of mutations based upon the directionality of the replication fork movement in the mutated region. Applying this analysis to CLAPS-seq reads in KBrO₃-treated WT cells revealed an equal distribution of 8-oxoG lesions on the leading and lagging template strands (Fig. 6A). KBrO₃-induced mutations however displayed slightly more G > T substitutions on the leading strand. While statistically significant ($p = 0.0302$ by two-sided paired t-test comparing the number of G > T substitutions on the leading and lagging strands in individual samples normalized to the strand specific sequence composition), the leading strand mutational bias induced by KBrO₃ treatment was only 9.35% that of other replication-associated mutagenic processes, like APOBEC signature mutations (Supplementary Fig. 17A), suggesting that any preferential removal of 8-oxoG from the lagging strand template by DNA repair (potentially MMR) is limited. HMCES deletion exacerbated the 8-oxoG leading strand bias ($p = 1.08×10^{-4}$ by two-sided paired *t*-test), indicating that HMCES may favor bypass of leading strand lesions. Interestingly, loss of Pol η removed the replication strand asymmetry. This indicates that Pol η likely mediates error-free bypass of 8-oxoG in the lagging strand template in human cells. Similar results indicate Pol η TLS functions preferentially during lagging strand synthesis for bypass of UV photoproducts in human melanomas and fibroblasts[57], suggesting a general lagging strand preference for this TLS polymerase. Similar mild leading strand bias was observed for G > T substitutions in untreated *MUTHY⁻/⁻* and *OGG1⁻/⁻* neuroblastoma cells (Supplementary Fig. 17B) that are defective in 8-oxoG repair, indicating a general better bypass of 8-oxoG on the lagging strand independent of the chemical species creating the lesion or cell type context.

We also assessed whether KBrO₃-induced mutations displayed transcriptional asymmetry, which would be indicative of transcription coupled repair of 8-oxoG. Transcription-coupled NER and BER have been suggested to be involved in 8-oxoG removal[58,59]. However, little transcriptional mutation asymmetry has been reported for ROS-associated SBS18 mutations[10], suggesting that transcription coupled repair of 8-oxoG may be limited. We found that G > T substitutions in WT and *HMCES⁻/⁻* cells only slightly favored the transcribed strand of genes (Fig. 6B). This bias was also observed in CLAPS-seq reads for 8-oxoG lesions, suggesting this effect is caused by a preference in lesion formation instead of a repair process. Transcriptional asymmetry in KBrO₃-treated Pol η-deficient cells, however, was very pronounced favoring the non-transcribed strand ($p$-value $< 1×10^{-5}$ by Chi-square), which impacted all G > T, G > C, and G > A substitution types. The effect size of this asymmetry is on par with other mutational processes limited by transcription-coupled nucleotide excision repair[56] (Supplementary Fig. 18A) and provides strong evidence that transcription-coupled repair can remove 8-oxoG lesions but its impact in mutational data is obscured by the error-free bypass of the lesion by Pol η. Interestingly, increased mutation of the non-transcribed strand was also observed for G > T substitutions in *OGG1⁻/⁻* neuroblastoma cells (Supplementary Fig. 18B), indicating that TC-NER can also remove 8-oxoG to help compensate for the absence of BER.

## Discussion

### Mutations from endogenously and exogenously derived 8-oxoG

We observed that long-term treatment of RPE-1 cells with the oxidant KBrO₃ increases substitutions and produces a mutational signature like COSMIC SBS18 that is proposed to be caused by endogenous ROS. Both signatures are composed of C > A mutations, however, the preferred trinucleotide sequences that these substitutions occurred in differ. Specifically, KBrO₃ exposure produced an over-representation of substitutions at CCA and CCT and under-representation at GCA, and GCT compared to SBS18. Other studies have demonstrated the same KBrO₃ signature for both long term exposure in RPE-1 cells[50] or short term exposure of human iPSCs[60], indicating that differences in exposure protocol and/or cell lines are not responsible for the differences in

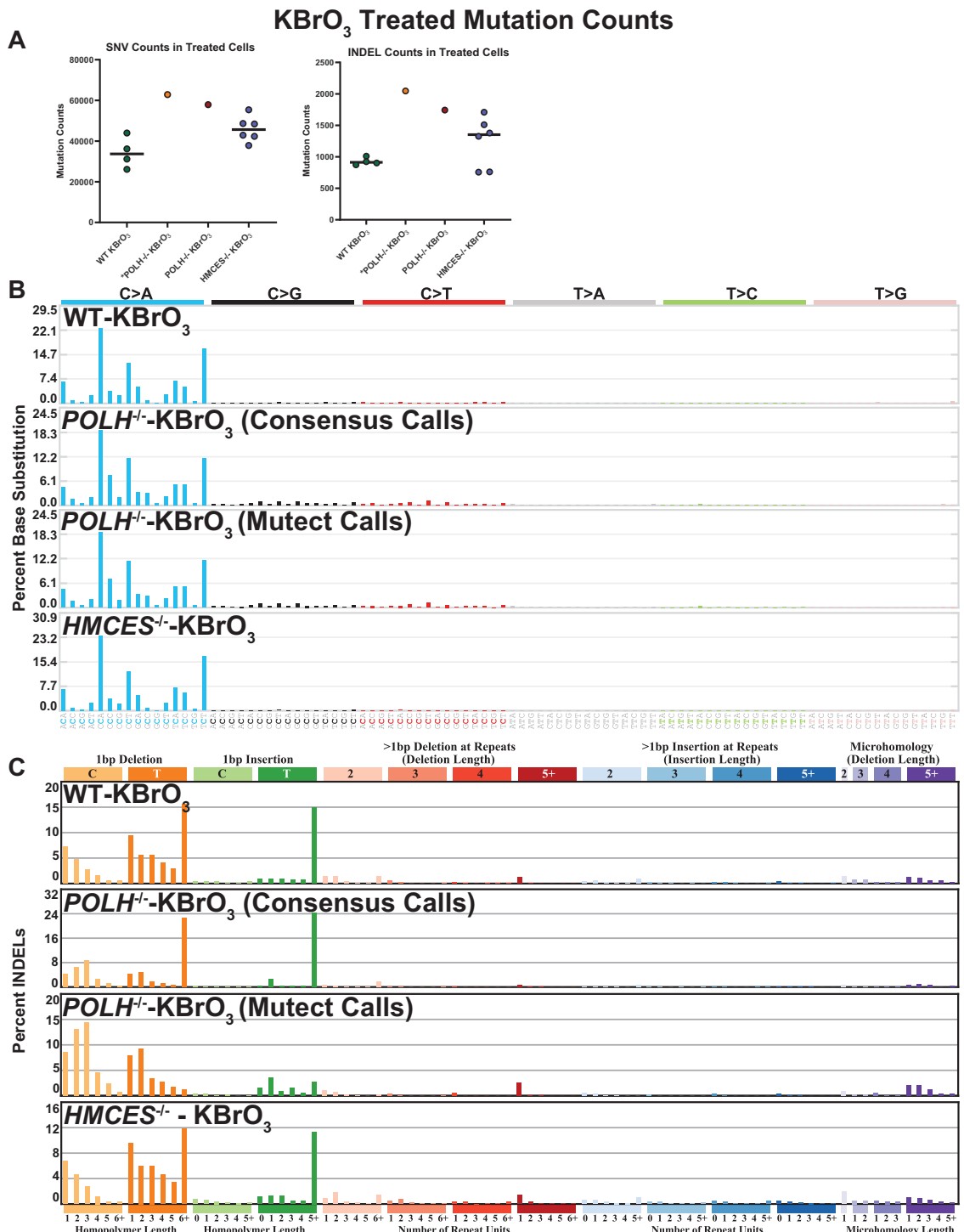

**Fig. 5 | KBrO₃-induced mutagenesis in human WT, *POLH*⁻/⁻, or *HMCES*⁻/⁻ RPE-1 cells lines. A** Number of SNVs and INDELs per genome in KBrO₃-treated for WT (*n* = 4), POLH⁻/⁻ (*n* = 1), and *HMCES*⁻/⁻ (*n* = 6) RPE-1 cells lines. Circles indicate independent genomes sequenced and horizontal bars are median values. Raw sequencing of the *POLH*⁻/⁻ cell line obtained from[50] was also reprocessed (indicated with an *) using BWA-mem and the consensus variant calling pipeline employed for the WT and *HMCES*⁻/⁻ cells. KBrO₃-associated **B** SNV and **C** INDEL mutation signatures from treated genomes.

SBS18 and KBrO₃-induced substitution specificity. Additionally, CLAPs-seq mapping of KBrO₃-induced 8-oxoG lesions in HeLa cells produced similar sequence preferences as KBrO₃-induced mutations, providing evidence that the oxidizing agent can dictate the sequences most likely to form 8-oxoG. Interestingly, CCA and CCT mutation contexts correspond to trinucleotides with low vertical ionization potential (VIP), which sensitizes these motifs (i.e. TGG and AGG sequences) to long-range guanine oxidation by charge transfer[61]. Reciprocally, TGC and

AGC have higher VIPs, indicating that GCA and GCT sequences would have fewer mutations produced by this mechanism. This correlation suggests that KBrO₃ may induce more guanine oxidation through charge transfer than endogenous ROS, leading to a KBrO₃ specific 8-oxoG mutation pattern. An alternative possibility is that specific oxidants produce 8-oxoG at different sequences. While the mechanism by which KBrO₃ generates 8-oxoG is unknown, its requires the presence of a reducing agent, like glutathione, and is insensitive to

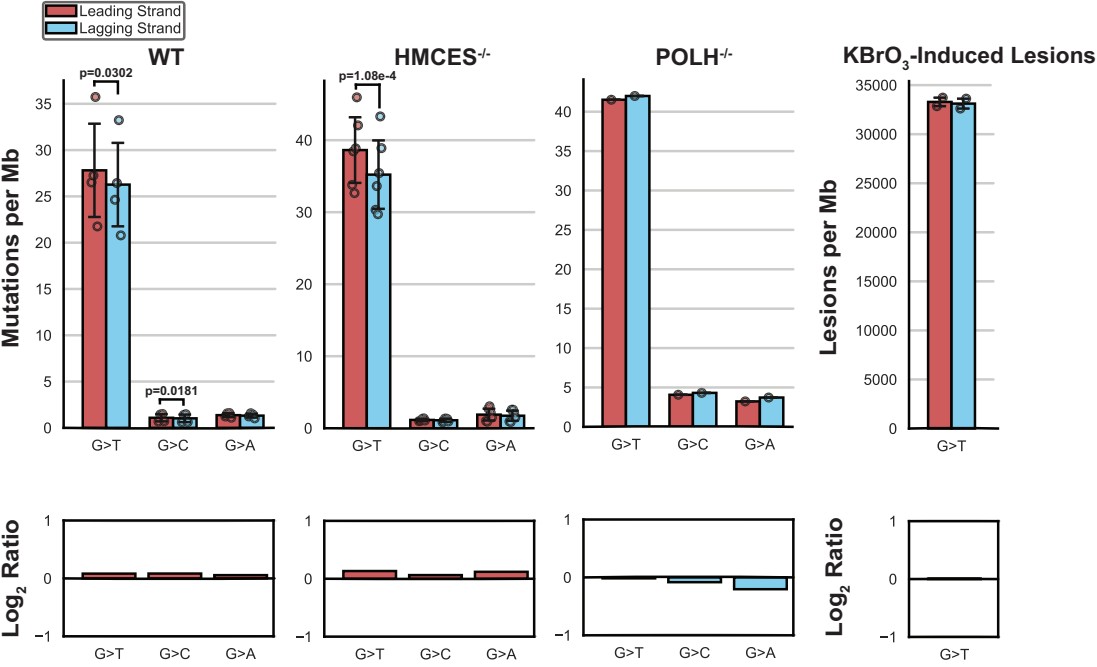

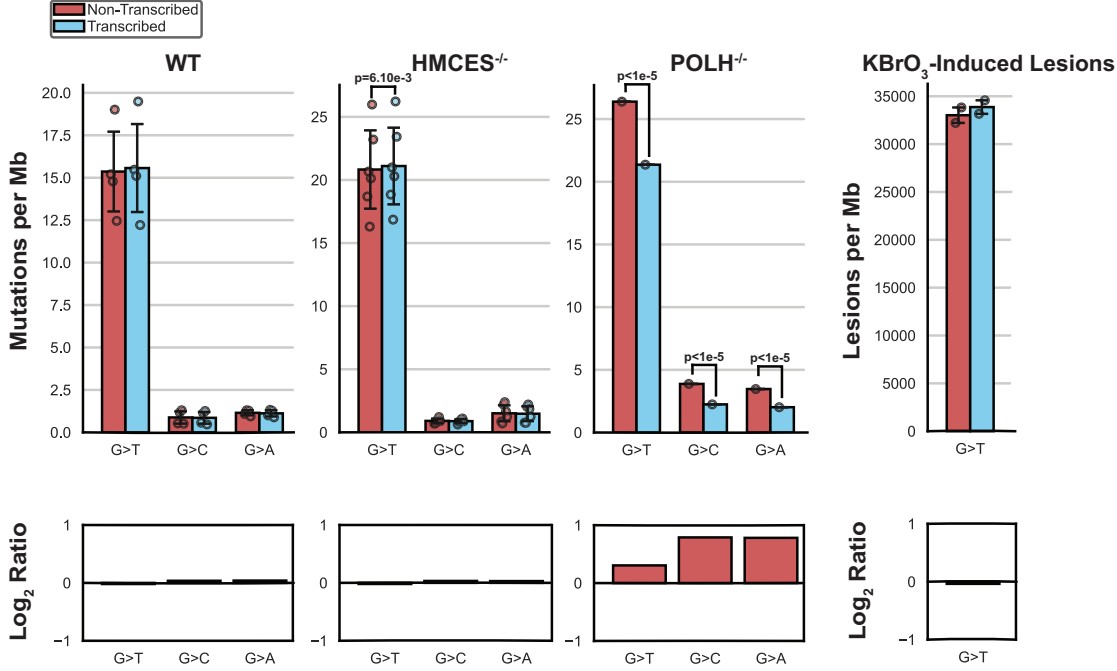

**Fig. 6 | KBrO₃-induced 8-oxodG mutation and lesion strand bias on leading/lagging and transcribed/non-transcribed strands in human cells. A** Mean values of G > T, G > C, and G > A mutations on the leading strand (red bars) or on the lagging strand (blue bars). **B** Mean values of G > T, G > C, and G > A mutations on the non-transcribed strand (red bars) transcribed strand (blue bars). circles represent values from each biologically independent sequenced genome ($n = 4$, 6, and 1 for WT, *HMCES*[-/-], and *POLH*[-/-], respectively) or technical replicate measurements of KBrO₃-induced lesions ($n = 2$). The bar graph below each plot represents the log₂(red/blue) value of the for the event per Mb bars. Significant differences between strands are indicated with *p*-values determined by two-sided paired *t*-test for WT and *HMCES*[-/-] genomes. *p*-values (calculated using GraphPad Prism) for G base mutation types in the *POLH*[-/-] sample were calculated by two-sided Chi-square against the null hypothesis of equal representation of mutations on both DNA strands.

traditional cellular ROS scavengers, indicating a different oxidation chemistry than for endogenous ROS[4]. By extension, the DNA damage induced by endogenous ROS could result from the combined activity of multiple different species (e.g. peroxide, superoxide, etc.), which may all have different sequence preferences in forming 8-oxoG. In the future, utilizing human cell systems to determine the mutation signatures of individual endogenous reactive oxygen species will be beneficial in determining which sources of ROS are most relevant for inducing mutation in human cancers.

KBrO$_3$ exposure primarily produces DNA damage in the form of 8-oxoG[35], which canonically produces G > T substitutions through Hoogsteen base pairing of 8-oxoG with dA[62] during DNA synthesis. We were therefore surprised to observe that KBrO$_3$ treatment also increased INDEL mutations and that a large percentage of these mutations occurred at A:T pairs. While a 1 bp deletion of C bases could logically stem from error-prone replication past an KBrO$_3$-induced 8-oxoG, the presence of a similar number of 1 bp T base deletions suggested that these mutations were caused either through collateral mutagenesis[36] adjacent to an 8-oxoG or by a second KBrO$_3$-induced DNA lesion. T deletions lacked an enrichment of C:G base pairs flanking the mutation, indicating that they were unlikely incurred as collateral mutations during 8-oxoG bypass. The lack of a reasonable connection of T deletions to 8-oxoG suggests that KBrO$_3$ also causes at least one other mutagenic DNA lesion and targets T or A bases, such as thymine glycol[4,63]. Our analysis of INDEL signatures in human tumors displaying high levels of SBS18 mutations indicated that a similar INDEL process occurs in these tumors. Moreover, the number of SBS18 mutations correlates with the number of mutations attributed to our KBrO$_3$-induced INDEL signature and the similar COSMIC ID5 signature, both suggesting that endogenous ROS produces these mutations. Additional research is needed to identify the specific DNA lesions causing these signatures.

## Oxidation-induced mutagenesis within the context of chromatin

Chromatin structure is another major influence on the density of 8-oxoG-induced mutations. Higher mutation densities were observed in more compact regions of the genome, which likely stems from the density of nucleosomes within these regions. Within nucleosomes, mutations had a 10.3 bp periodicity when treated with KBrO$_3$, occurring primarily on nucleotides proximal to the histone octamer. This suggests that DNA repair mechanisms may be excluded from the inward facing positions of the nucleosomal DNA. This is in stark contrast to UV-induced cyclo-pyrimidine dimer positioning at outward facing nucleotides at nucleosomes due to preferential lesion formation at these sites[64]. Prior work was unable to obtain a cryo-EM structure of OGG1 engaged with an 8-oxoG in the nucleosome[65], which was hypothesized to result from an inability of OGG1 to flip the 8oxoG embedded in the nucleosome into the enzyme active site. However, our structural data and another recently published OGG1-8-oxoG-NCP structure[66] clearly demonstrate that OGG1 accesses outward facing 8-oxoG in the nucleosome by sculpting nucleosomal DNA and flipping the 8-oxoG base into its active site for catalysis, similar to the DNA sculpting mechanisms previously observed for the DNA glycosylase AAG[67] and APE1[68]. This leaves inward facing 8-oxoG in the nucleosome more prone to mutation, as OGG1 lacks the ability to recognize 8-oxoG in these positions without massive changes in nucleosome structure[46,47]. Repair of these sites is likely significantly delayed and may require active nucleosome remodeling in response to DNA damage by additional cellular factors, such as the BER-associated nucleosome remodeler ALC1[69]. Interestingly, other protein-DNA interactions appear to have little to no impact on 8-oxoG-induced mutagenesis. Transcription factors bound to gene promoter regions produce no change in the density of KBrO$_3$-induced mutations in contrast to other types of DNA damage, like cyclobutane pyrimidine dimers. 8-oxoG lesions at these sites are greatly reduced compared to neighboring DNA, suggesting that they are rapidly repaired. However, we are unable to exclude the possibility that transcription factor binding protects their binding sites from formation of 8-oxoG. In contrast to the variable sequence preferences for 8-oxoG formation induced by KBrO$_3$ or ROS, chromatin impacts on 8-oxoG induced mutation appear to be largely conserved for each method of lesion formation as the observed distributions of mutation with respect to chromatin state, nucleosome occupancy, replication and transcriptional strand bias, and transcription factor binding sites for KBrO$_3$-induced mutation largely mirror those reported for SBS18 mutations in human tumor genomes[10,30].

## An expanded 8-oxoG repair network in human cells

Our data indicates at least three mechanisms of 8-oxoG repair influence its mutagenicity (Fig. 7). OGG1 and MUTYH activities provide the first line of defense against 8-oxoG mutagenesis as their deletion results in spontaneous mutator phenotypes displaying SBS18-like mutation spectra. 8-oxoG escaping BER can be bypassed by the replication-associated damage tolerance mechanisms of Pol η and HMCES. The 1.5-fold increase in mutation load in KBrO$_3$-treated Pol η-

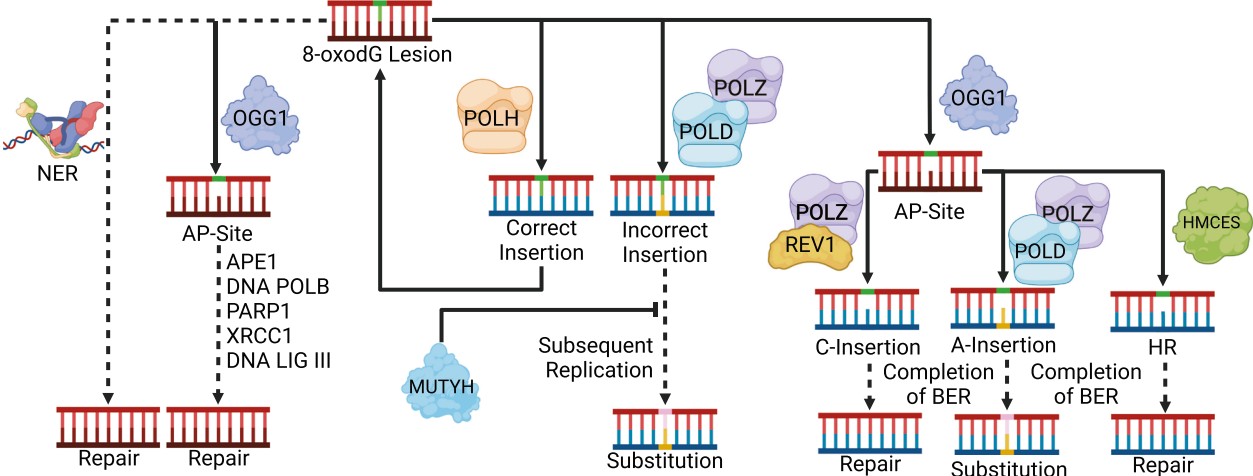

**Fig. 7 | Mechanisms that limit KBrO$_3$-induced 8-oxodG mutagenesis in human cells.** Mutations from 8-oxoG can be limited in human cells by OGG1- and MUTYH-initiated BER. Secondary limits on mutagenesis include Pol η, HMCES, and transcription-coupled repair. Dashed lines represent multi-step processes. Image created with BioRender.com under agreement number DX26OBAXYI.

deficient cells is significantly lower than the impact of Pol η loss in yeast[25]. This suggests either lesser reliance on Pol η due to redundancy with other human DNA polymerases in 8-oxoG bypass, or less accurate Pol η bypass in humans compared to yeast. Supporting the latter, biochemical experiments have shown human Pol η correctly inserts dC across from 8-oxoG to a lesser extent than yeast Pol η[25]. HMCES-deficiency also resulted in elevated $KBrO_3$-induced mutagenesis, consistent with the previously reported sensitivity of $HMCES^{-/-}$ cells to $KBrO_3$[51]. $KBrO_3$-treated HMCES-deficient cells displayed a mutation spectrum identical to $KBrO_3$-treated wildtype cells suggesting that HMCES aids in error-free bypass of 8-oxoG or derived repair intermediates, despite its previously demonstrated role preventing mutagenesis[51,55,70,71] by cross-linking to AP-sites[51]. Such a mutation spectrum could be reconciled with HMCES's known biochemical activity toward AP-sites if OGG1 removal of 8-oxoG can be uncoupled from the rest of BER, resulting in AP-sites occurring specifically at dG nucleotides. Subsequent TLS-based bypass of these AP-sites could either be error-free (in the case of REV1-mediated bypass via C-insertion) or produce C > A substitutions (by A-rule AP-sites site bypass), which would recapitulate the $KBrO_3$ substitution signature. The error-prone bypass of AP-sites may also cause the oxidation-induced 1 bp deletions we identified and that are elevated in $HMCES^{-/-}$ cells[72]. By analogy with Ung1 generated AP-sites sites from dU in the lagging strand template[73], uncoupling of OGG1 glycosylase activity from BER would be expected to occur more frequently at sites of DNA replication (where HMCES functions) as the glycosylase could recognize 8-oxoG in ssDNA, but subsequent steps of BER would be inhibited do to the lack of a complementary DNA strand. $KBrO_3$-treated $HMCES^{-/-}$ cells also displayed enhanced leading strand replication strand-bias compared to similarly treated WT cells suggesting HMCES may function more during leading strand synthesis. The underlying reason for this preference is currently unclear, especially considering the synthetic lethal phenotype between HMCES-deficiency and APOBEC3A expression[53,74], which damages the lagging strand template[75]. Transcriptional strand asymmetries in $KBrO_3$-treated Pol η−deficient cells and untreated $OGG1^{-/-}$ cells ultimately revealed a substantial decrease in the mutational burden on the transcribed strand, suggesting that TC-NER can remove oxidative damage when other repair processes are overwhelmed or defective. Previous studies indicate that CSB, a major component of TC-NER, can be recruited to transcription sites upon oxidative damage and is important for fitness following oxidative damage[59,76].

In conclusion, this study outlines the multi-dimensional mutational landscape of exogenously and endogenously induced oxidative damage and the consequences of topology on this landscape while providing mechanistic insight into primary, secondary, and tertiary strategies to limit 8-oxoG mutagenesis in human cells. Variants and loss of multiple of the factors in this study lead to cancer and neurodegenerative disease including OGG1 and Pol η. The robust bioinformatics pipeline and exhaustive topological analysis can also be used as a blueprint and foundation to develop a database of holistic multidimensional mutational signatures to explore mechanism and drug targets. Future research should explore topological mutagenesis studies of various types of damage and genetic conditions in human cells. This research should also aim to understand the interplay between these pathways and identify potential therapeutic targets for future interventions.

## Methods
### Cell Culture
hTERT-RPE-1 p53$^{-/-}$ cells were cultured in DMEM (Thermo Fisher Cat No. 11965092 supplemented with 7.5% FBS, 1X Glutamax (Thermo Fisher Cat No. 35050061), 1X Non-Essential Amino Acids (Thermo Fisher Cat No. 11140035, and 1X Penicillin-Streptomycin (Thermo

Fisher Cat No. 15140122) at 37° C in 5% CO2. Cells were single-cell cloned with cloning rings. Single-cell clone parental cells were transfected with pSpCas9(BB)-2A-Puro 2 (Addgene Cat No. 48139) that contain guide RNAs that target the intron-exon junction of the second exon of *HMCES* (5'-TTGCGCCTACCAGGATCGGC and 5'-AC TTTAGACGGTGGTCACGG). Cells were selected with 15 µg/mL puromycin for two days prior to plating for individual clones because hTERT-RPE-1 p53$^{-/-}$ are already mildly puromycin resistant. Clones were screened for deletion by PCR (using primers HMCES F2: 5'-GC ATTTGCAGAGCTCCTCTC and HMCES R2: 5'-GACAGAAGCACTGG GCTG) and by immunoblotting for loss of HMCES expression with antibodies raised against the middle and C-terminus of the protein. $HMCES^{-/-}$ cells were confirmed to be hypersensitive to $KBrO_3$, consistent with previous results[51].

### Measurement of cellular ROS
For the reactive oxygen species assay, a single-cell clone (Clone 3) of WT hTERT-RPE-1 p53$^{-/-}$ was seeded into an optically clear 96-well plate. 24 hours later cells were treated with 250 µM $KBrO_3$ or vehicle for an hour, followed by treatment with 5 µM of CellRoX 488 reagent and NucBlue for 30 minutes. Cells were fluorescently imaged live with a Nikon Ti2E microscope and analyzed for integrated nuclear fluorescent intensity using Nikon NIS Elements. CellRoX488 fluorescence intensities were plotted using GraphPad Prism.

### Long-term Mutagenesis Assay
For long-term mutagenesis assays, pooled parental and a single-cell clone (Clone 3) of WT hTERT-RPE-1 p53$^{-/-}$ (generously provided by Daniel Durocher, University of Toronto) as well as three individual single-cell $HMCES^{-/-}$ clones of HMCES knockouts (Clone3.1, Clone 3.3, and Clone 3.4) were seeded into 10 cm dishes and carried continuously in the presence or absence of 250 µM $KBrO_3$ for 100 generations (3 months). Each passage, cells were seeded at similar cell numbers (20% confluency) and carried until 80% confluent at which point they were passaged again. After 100 generations (24 passages), each cell line was single-cell cloned and two of each clone (WT pool, WT Clone 3, $HMCES^{-/-}$ Clones 3.1, 3.3, 3.4) were harvested for genomic DNA (Promega Cat No. A1120). Genomic DNA was submitted for 150 bp paired-end Illumina dep-sequencing sequencing targeting 30X depth at Vanderbilt University's VANTAGE Next Generation-Sequencing core.

### System Information
All computational analyses were performed on Linux, specifically Ubuntu 22.04.03 LTS. Data analysis was conducted using Python v3.10.12, Python v2.7.18, Perl v5.34.0 and R v4.1.2 (unless a virtual environment was required). Further system, software, library versions, and hardware information is available on request.

### Sequencing Alignment
Results were aligned to the Genome Reference Consortium Human Build 37 (GRCh37/hg19) using the Burrows-Wheeler Aligner (BWA) **mem** algorithm on default parameters (BWA v0.7.17). The resulting Sequence Alignment/Map (SAM) files, which contain aligned sequence reads, were compressed into Binary Alignment/Map (BAM) format using **samtools view** (samtools v1.13 using htslib v1.13+ds). [Note: All **samtools** steps were run using default parameters to maintain a standard approach] After compression, the BAM files were sorted based on genomic coordinates using **samtools sort** to prepare for removal of duplicate reads which can arise from PCR amplification artifacts during sequencing. These were removed using **samtools rmdup** so it would not have an impact on downstream variant calling and analysis. These final BAM files were converted to MPILEUP files using **samtools mpileup**. The final BAM and MPILEUP files were used to call mutations from multiple mutation callers.

## Mutation Calling

The BAM files were processed with Strelka2 (v2.9.10), Manta (v1.6.0), and Somatic Sniper (v1.0.5.0) while the MPILEUP files were processed using VarScan2 (v2.3). SNVs and INDELs were called using **VarScan somatic** comparing treated cells to untreated counterparts with the following parameters changed **-min-coverage 10 -min-var-freq 0.2 -somatic-_p_-value 0.05 -min-freq-for-hom 0.9 -min-avg-qual 30** to reduce artifacts of mutation calling. The resulting SNV and INDEL files were split into germline, somatic, and loss of heterozygosity (LOH) files using **VarScan processSomatic** on default parameters, to split the results and isolate the high confidence somatic SNV and INDEL mutation calls which were used to identify consensus mutations.

The BAM files were initially compared to their corresponding normal counterparts utilizing Manta's structural variant pipeline[77], employing default parameters to detect small INDEL candidates for input into Strelka2. Strelka2 was run on default parameters comparing tumor to normal using hg19 and Manta's INDEL candidates for the tumor/normal pair. The resulting SNV and INDEL mutation calls were used to identify consensus mutations. The BAM files were also used to create a third set of SNV calls using Somatic Sniper on default parameters except **-Q 40 -G -L** which requires a minimum somatic score of 40 as recommended by the developers for BWA aligned reads, and not report loss of heterozygosity (LOH) and gain of reference (GOR) mutations in the final output to reduce the likelihood of false positives. The resulting SNV mutation calls were also used to identify consensus mutations.

To account for artifacts of mutation calling and sequencing from different callers, we took the consensus from all three callers (Strelka2, Somatic Sniper, and VarScan2) for SNVs and the consensus from both Strelka2 and VarScan2 for INDELs. This was done using a custom Python script requiring mutations to be present in all sets of mutation calls for the sample. Then all the separate consensus mutations were pooled, and mutations present in more than one sample were omitted due to a high likelihood of being a germline mutation or artifact of sequencing and mutation calling. The concatenated mutation calls were then split into separate sets based on treatment, genotype, or both depending on the analysis.

## Processing of 8-oxoG Lesion Data

CLAPS-seq FASTQ files were aligned to hg19 using the **bwa-mem** algorithm on default parameters. The resulting SAM files were processed using a custom Python script to convert the SAM file into a BED file. The script filtered out reads that did not align with a CIGAR score of 150 M. It also filtered reads keeping ones that aligned to chromosome 1-22, X, or Y. It took the reads passing this filter and checked the bitwise flag for 0 (complemented) or 16 (reverse complemented) and processed the proper alignment position (either 5' or 3' of the top strand) to determine the base pair position where the lesion occurred. We then filtered the custom BED file for positions where there was a G at that context which removed reads which were assumed to be false positives reported by the authors. The resulting BED files were converted to a VCF format using a custom Python script to process this data through other programs, like vcf2maf and nucleosome profiling.

## Mutation Signature Generation

We generated mutation signatures from cell mutations using SigProfilerExtractor[34] (v1.1.23) on default parameters with a minimum and maximum of 1 and 5 signatures respectively. The most stable number of signatures KBrO$_3$-associated was 2 for both SNV and INDEL mutation signatures, which was used for all subsequent comparisons. All other signatures generated used the most stable number of signatures.

## Correlation of INDEL signature with SBS18

The PCAWG data was analyzed using the MutationalPatterns package in R[37] to conduct non-negative matrix factorization (NMF). This analysis incorporated the COSMIC SBS and INDEL signatures, along with an additional custom INDEL signature derived from the NMF results of KBrO$_3$ treated cells. Tumors were deemed positive for SBS18 and the custom INDEL signature if they exhibited a minimum of 20 mutations associated with each signature. These samples were plotted with the log$_2$ transformed number of mutations. The Pearson correlation coefficient was computed based on the mutation count per sample for each signature. The SigProfiler signatures in samples data provided with PCAWG were used for the correlation of ID5 with SBS18.

## Strand Asymmetry

Replication strand asymmetry was calculated similarly to AsymTools2[56] on default parameters. Custom Python scripts were generated to calculate replication strand asymmetry for individual samples using map of left or right replicating regions in the hg19 reference genome provided within AsymTools2. Mutational strand preference among replicate samples was assessed by two-sided paired t-test. To produce a replication asymmetry of APOBEC-induced mutation in tumors for comparison to 8-oxoG mutational asymmetry, C to G substitutions in TCW contexts of BRCA-proficient breast cancers were filtered from ICGC mutational data as described in[78]. Transcribed strand asymmetry was calculated using a custom Python script using a similar approach to AsymTools2. The script takes an RPE-1 transcribed gene list from GEO accession number GSE146121 and cross-references the list with the UCSC hg19 gene list. This provided us with a gene list that was actively transcribed in RPE-1 cells, which we then compared with mutations and lesions. Mutations mapping to the top strand with a G base were considered to be on the (+) strand and mutations mapping with a C base were considered to be on the (−) strand. By analyzing the gene's orientation, we were able to ascertain the strand on which the event took place and subsequently compare the occurrences of each event on both the transcribed and non-transcribed strands. To normalize the events, the event counts were divided by the guanine base count on that transcribed or non-transcribed strand, respectively, resulting in the unit of events/Mb. The results were similar to what was represented in the AsymTools2 results, however, were specific to the cell line and had a higher resolution since transcribed regions were not binned but were measured at single-nucleotide resolution.

## Chromatin State & Nucleosome Profiling

Chromatin states were assessed by mapping mutations and lesions to chromatin states from the epithelial cell HMM chromatin maps (https://genome.ucsc.edu/cgi-bin/hgFileUi?db=hg19&g= wgEncodeBroadHmm) using bedtools intersect[79,80]. Subsequently, we standardized the results to events/Mb based on the HMM map's region size. The order of heterochromatin to euchromatin was determined by the map construction.

Mutations and lesions were intersected with strongly positioned nucleosome dyads following the protocol outlined in[81] in a 1000 base-pair (bp) window. Expected counts were calculated using genomic trinucleotide mutation or lesion frequencies multiplied by the occurrence of those contexts at each position in the dyad map. The observed counts were divided by the expected counts and log$_2$ transformed to generate the graphs.

The data was smoothed using a Savitzky–Golay filter with a 200 bp window with a polynomial order of 3.

$$\log_2\left(\frac{actual\ number\ of\ AAA\ evebts\ at\ map\ postition + \ldots + actual\ number\ of\ TTT\ events\ at\ map\ position}{(genomewide\ AAA\ event\ frequency * AAA\ context\ at\ map\ position) + \ldots + (genomewide\ TTT\ event\ frequency * TTT\ context\ at\ map\ position)}\right)$$

## Transcription Factor Profiling

Mutations and lesions were intersected with known active transcription factor binding sites using a map generated from previous work[45] in a 1000 bp window. Events were counted and graphed using a custom Python script and smoothed using a Savitzky−Golay filter with a 200 bp window with a polynomial order of 3.

## Purification of H. sapiens OGG1 K249Q

A pGEX6P1 vector (N-terminal GST tag) with the *H. sapiens* OGG1 gene bearing the K249Q mutation was obtained from GenScript. For protein expression, the pGEX6P1-OGG1-K249Q vector was transformed into BL21-CondonPlus (DE3) RIPL cells (Agilent). The transformed cells were grown in 2x YT media at 37 °C until an $OD_{600}$ of 0.8 and protein expression induced with 0.5 mM IPTG overnight at 18 °C. The cells were harvested by centrifugation and resuspended in a buffer containing 50 mM HEPES (pH-7.5), 150 mM NaCl, 1 mM DTT, and a protease inhibitor cocktail (Benzamidine, Leupeptin, AEBSF, Pepstatin A). The resuspended cells were lysed by sonication and the lysate clarified by centrifugation. The clarified lysate was loaded onto a GSTrap HP column (Cytiva) equilibrated with 50 mM HEPES (pH-7.5), 150 mM NaCl, and 1 mM DTT, and the protein was eluted in a buffer containing 50 mM HEPES (pH-7.5), 150 mM NaCl, 1 mM DTT, and 50 mM reduced glutathione. Fractions containing GST-OGG1 were loaded onto a Resource S cation exchange column (Cytiva) equilibrated with 50 mM HEPES (pH-6.8), 50 mM NaCl, 1 mM DTT, and 1 mM EDTA, and eluted in a high salt buffer containing 50 mM HEPES (pH-6.8), 1 M NaCl, 1 mM DTT, and 1 mM EDTA. OGG1 was then liberated from the GST-tag by incubation with PreScission Protease for 4 hours in a buffer containing 50 mM HEPES (pH-7.5), 150 mM NaCl, and 1 mM DTT. The cleaved OGG1 protein was rerun over a Resource S cation exchange column (Cytiva), and the eluted protein loaded on a Sephacryl S-200 HR (Cytiva) equilibrated with 50 mM HEPES (pH-7.5), 150 mM NaCl, and 1 mM TCEP. The purified OGG1 fractions were combined, concentrated to 10 mg ml$^{-1}$, and stored at -80 °C.

## Preparation of oligonucleotides

DNA oligonucleotides (oligos) containing 8-oxoG were obtained from TriLink BioTechnologies, and non-damaged oligos were obtained from Integrated DNA Technologies. Each oligo was resuspended at 1 mM in a buffer containing 10 mM Tris (pH-8.0) and 1 mM EDTA. Complimentary oligos (see Supplementary Table 2) were mixed at a 1:1 ratio and annealed by heating to 90 °C followed by a stepwise cooling to 4 °C using a linear gradient at -1 °C min$^{-1}$. The annealed oligos were stored long-term at -20 °C.

## Purification of recombinant human histones

The genes encoding *H. sapien* histones H2A, H2B, H3.2 (C110A), and H4 were cloned into a pet3a expression vector. For histone H2A, H3.2, and H4 expression, vectors were transformed into T7 Express lysY competent cells (New England Biolabs). For histone H2B expression, the vector was transformed into BL21-CodonPlus (Agilent) competent cells. The cells were grown in minimal media at 37 °C until an $OD_{600}$ of 0.4 was reached, and protein expression induced with 0.4 mM IPTG (H2A, H2B, and H3.2) or 0.3 mM IPTG (H4) for 3-4 hours at 37 °C. The cells were harvested by centrifugation and resuspended in a buffer containing 50 mM Tris (pH-7.5) 100 mM NaCl, 1 mM benzamidine, 1 mM DTT, and 1 mM EDTA. The histones were purified under denaturing conditions using an established method[82,83]. In brief, the resuspended cells were lysed by sonication, inclusion bodies isolated by centrifugation, and the histones extracted from the inclusion bodies under denaturing conditions (6 M Guanidinium chloride). After extraction, the histones were purified using subtractive anion-exchange chromatography and cation-exchange chromatography using gravity flow columns. The purified histones were then dialyzed into $H_2O$, lyophilized, and stored at -20 °C.

## Preparation of H2A/H2B Dimers and H3/H4 Tetramers

H2A/H2B dimers and H3/H4 tetramers were prepared using an established method[82,83]. In brief, each individual histone was resuspended in a buffer containing 20 mM Tris (pH-7.5), 6 M guanidinium chloride, and 10 mM DTT. For H2A/H2B dimers, H2A and H2B were mixed at a 1:1 ratio and dialyzed three times against a buffer containing 20 mM Tris (pH-7.5), 2 M NaCl, and 1 mM EDTA. For H3/H4 tetramers, H3 and H4 were mixed at a 1:1 ratio and dialyzed three times against a buffer containing 20 mM Tris (pH-7.5), 2 M NaCl, and 1 mM EDTA. The H2A/H2B dimers and H3/H4 tetramers were subsequently purified over a Sephacryl S-200 HR (Cytiva) in a buffer containing 20 mM Tris (pH-7.5), 2 M NaCl, and 1 mM EDTA. The purified H2A/H2B dimers and H3/H4 tetramers were stored in 50% glycerol at -20 °C.

## Nucleosome assembly and purification

Recombinant nucleosomes were assembled by an established salt-dialysis method[82,83]. In brief, H2A/H2B dimers and H3/H4 tetramers were mixed with DNA in a 2:1:1 molar ratio, respectively, in a buffer containing 20 mM Tris (pH 7.5), 2 M NaCl, and 1 mM EDTA. Stepwise nucleosome assembly was then performed by decreasing the amount of NaCl from 2.0 M NaCl to 1.5 M NaCl, 1.0 NaCl, 0.66 M NaCl, 0.5 M NaCl, 0.25 M NaCl, 0.125 M, and 0 M NaCl over a period of 24 - 26 hours. The reconstituted nucleosomes were heat shocked at 37 °C for 15 minutes to generate uniform DNA positioning and purified by ultracentrifugation over a 10% - 40% sucrose gradient. Final nucleosome purity was determined using native polyacrylamide gel electrophoresis (5%, 59:1 acrylamide:bis-acrylamide), and the purified nucleosomes were stored at 4 °C.

## Cryo-EM sample and grid preparation

For cryo-EM sample preparation, 8-oxoG-NCP (5 μM) was mixed with OGG1 K249Q (7.5 μM - 10 μM) in a buffer containing 25 mM HEPES (pH-7.1), 25 mM NaCl, 1 mM TCEP, and 1 mM EDTA. The OGG1-8-oxoG-NCP complexes were then incubated at 4 °C for 10 minutes and fixed with glutaraldehyde (0.1%) for 20 minutes. The samples were loaded onto a Superdex S200 Increase 10/300 GL (Cytiva) equilibrated with a buffer containing 50 mM HEPES (pH-7.1), 100 mM NaCl, 1 mM TCEP, and 1 mM EDTA. Fractions containing OGG1-NCP were identified via native polyacrylamide gel electrophoresis (5%, 59:1 acrylamide:bis-acrylamide). The fractions containing the OGG1-8-oxoG-NCP complex were then combined and concentrated to 1.5 μM for short-term storage. Gels corresponding to the 8-oxoG-NCP − 6 and 8-oxoG-NCP + 4 samples used for cryo-EM grid preparation can be found in Supplementary Fig. 5A and 8 A. The samples (3 μL, 1.5 μM) were then applied to a Quantifoil R2/2 300 mesh copper cryo-EM grid at 8 °C and 95% humidity, and the grids plunge frozen in liquid ethane using a Vitrobot Mark IV (Thermo Fisher).

## Cryo-EM Data collection and processing

All cryo-EM data collections were performed on a Titan Krios G3i equipped with Gatan K3 direct electron detector and BioContinuum energy filter at the University of Chicago Advanced Electron Microscopy Core Facility (RRID:SCR_019198). All cryo-EM datasets were

processed with cryoSPARC[84] using the workflows outlined in Supplementary Fig. 5 and 9. In brief, the micrographs were corrected for beam-induced drift using Patch Motion Correction and contrast transfer function (CTF) fit using Patch CTF Estimation. The micrographs were then manually curated to exclude micrographs of poor quality. Following micrograph curation, a subset of micrographs was subjected to blob picking to generate initial templates, which were then used for automated template picking. The particle stacks were then extracted from the micrographs and multiple rounds of 2D classification were performed. Ab-initio models were then generated using the final particle stacks and several rounds of heterogeneous refinement performed to initially separate 8-oxoG-NCP and OGG1-8-oxoG-NCP maps.

To improve the interpretability of the 8-oxoG-NCP maps, additional 3D-classification was performed using a focus mask for the entry/exit site nucleosomal DNA, which is prone to partially unwrapping from the histone octamer. Following 3D classification, the final particle stacks for each 8-oxoG-NCP structure were re-extracted to full box size (600 pixels), and the re-extracted particles subjected to local CTF refinement and non-uniform refinement. The final 8-oxoG-NCP maps were then subjected to a B-factor sharpening using PHENIX autosharpen. The final 8-oxoG-NCP maps were deposited into the electron microscopy data bank under accession numbers EMD-43595 for 8-oxoG-NCP − 6 and EMD-43600 for 8-oxoG-NCP + 4.

To improve interpretability of the OGG1-8-oxoG-NCP maps, 3D-classification was performed using a focus mask for OGG1 and the surrounding nucleosomal DNA. Following 3D-classification, the final particle stacks for each OGG1-8-oxoG-NCP structure were re-extracted to full box size (600 pixels), and the re-extracted particles subjected to local CTF refinement and non-uniform refinement. To further improve interpretability of the maps, local refinement (without particle subtraction) was performed using a focus mask for OGG1 and the surrounding nucleosomal DNA or a focus mask for the NCP. A composite map for the OGG1-8-oxoG-NCP − 6 structure was then generated by combining the maps from a non-uniform refinement and two local refinement (OGG1/DNA and NCP local refine) jobs using PHENIX combine focused maps. A composite map for the OGG1-8-oxoG-NCP + 4 structure was then generated by combining the maps from the non-uniform refinement and local refinement (OGG1/DNA local refine) jobs using PHENIX combine focused maps. The final cryo-EM maps were deposited into the Electron Microscopy Data Bank under accession numbers EMD-43600 for OGG1-8-oxoG-NCP − 6 (composite), EMD-43597 for OGG1-8-oxoG-NCP − 6 (consensus), EMD-43598 for OGG1-8-oxoG-NCP − 6 (NCP local refine), EMD-43599 for OGG1-8-oxoG-NCP − 6 (OGG1/DNA local refine), EMD-43601 for OGG1-8-oxoG-NCP + 4 (composite), EMD-43602 for OGG1-8-oxoG-NCP + 4 (consensus), and EMD-43603 for OGG1-8-oxoG-NCP + 4 (OGG1/DNA local refine).

### Model building and refinement

All model building and refinement was performed iteratively using University of California San Francisco (UCSF) Chimera[85], PHENIX[86], and COOT[87]. An initial nucleosome model was generated using a previously determined cryo-EM structure of a nucleosome containing an AP-site (PDB: 7U52)[68]. The initial OGG1 model was generated from a previously determined X-ray crystal structure of an OGG1-8-oxoG-DNA complex (PDB:1EBM)[48]. The models for each respective structure were rigid body docked into the cryo-EM map using UCSF Chimera[85]. The models were then refined in PHENIX[86] using protein and nucleic acid secondary structure restraints, and manual adjustments to the models made in COOT[87]. All final models were validated using MolProbity[88], and model coordinates for each structure were deposited in the Protein Data Bank (PDB) under accession numbers 8VWS for 8-oxoG-NCP − 6, 8VWT for OGG1-8-oxoG-NCP − 6, 8VWU for 8-oxoG-NCP + 4, 8VWV for OGG1-8-oxoG-NCP + 4.

### Statistics & Reproducibility

The number of independent WT hTERT-RPE-1 p53[-/-] and hTERT-RPE-1 p53[-/-] *HMCES*[-/-] clones selected for sequencing was chosen to allow statistical comparisons in mutations per genome between genotypes and treatments by Mann-Whitney U test. Additionally, the length of passaging was chosen to acquire over 10,000 aggregate mutations in each treatment type based on previously established mutation rates of hTERT-RPE-1 cells in culture[89]. This number of mutations allows for robust statistical analysis comparing the density of mutations in different genome features. The number of analyzed *POLH*[-/-], *MUTYH*[-/-], and *OGG1*[-/-] clones was determined by the public availability of the sequencing data. No power calculation was used to predetermine sample size. No data were excluded from the analyses, the experiments were not randomized, and the Investigators were not blinded to allocation during experiments and outcome assessment.

### Reporting summary

Further information on research design is available in the Nature Portfolio Reporting Summary linked to this article.

## Data availability

The next generation sequencing data generated in this study for untreated and KBrO$_3$-treated hTERT-RPE-1 p53[-/-] and hTERT-RPE-1 p53[-/-] *HMCES*[-/-] cells have been deposited as FASTQ files at the NCBI SRA database under accession code PRJNA1100509. Full mutation lists used for analysis are provided in Supplementary Data 1. hTERT-RPE-1 *POLH*[-/-] VCF files used in this study are available from[50] available on Mendeley Data server (https://doi.org/10.17632/jkjkpvgxyd.1). FASTQ files for KBrO$_3$-treated hTERT-RPE-1 *POLH*[-/-] cells used I this study are available in NCBI SRA database under accession code PRJNA940340. *MUTYH*[-/-] and *OGG1*[-/-] VCF files can be obtained from the supplementary dataset S01 from[8]. CLAPS-seq 8-oxoG lesion mapping data from[39] are available from the Gene Expression Omnibus (GEO) (https://www.ncbi.nlm.nih.gov/geo/) under accession code GSE181312. Publicly available lists of tumor mutations were obtained from the International Cancer Genome Consortium (ICGC) from consensus_snv_indel/final_consensus_passonly.snv_mnv_indel.icgc.public.maf.gz and simple_somatic_mutation.open.BRCA-EU.tsv.gz. Corresponding tumor mutation lists can be downloaded from ICGC using the linked download instructions. The final cryo-EM maps are available from the Electron Microscopy Data Bank under accession numbers EMD-43600 for OGG1-8-oxoG-NCP − 6 (composite), EMD-43597 for OGG1-8-oxoG-NCP − 6 (consensus), EMD-43598 for OGG1-8-oxoG-NCP − 6 (NCP local refine), EMD-43599 for OGG1-8-oxoG-NCP − 6 (OGG1/DNA local refine), EMD-43601 for OGG1-8-oxoG-NCP + 4 (composite), EMD-43602 for OGG1-8-oxoG-NCP + 4 (consensus), and EMD-43603 for OGG1-8-oxoG-NCP + 4 (OGG1/DNA local refine). The model coordinates for each structure are available from the Protein Data Bank (PDB) under accession numbers 8VWS for 8-oxoG-NCP − 6, 8VWT for OGG1-8-oxoG-NCP − 6, 8VWU for 8-oxoG-NCP + 4, 8VWV for OGG1-8-oxoG-NCP + 4. All data is publicly available and accessible without restriction. Values underlying all graphs in figures are provided in the Source Data file. Source data are provided with this paper.

## Code availability

All custom scripts for mutation and lesion analyses[90] are available at the S-RobertsLab GitHub (https://github.com/S-RobertsLab/Cordero-et-al.-2024).

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

## Acknowledgements

Whole genome sequencing in this manuscript was conducted by the Vanderbilt Technologies for Advanced Genomics (VANTAGE) core. We thank Eric J. Crow for assistance establishing portions of our variant calling and mutation signature pipeline. This work was supported by the following National Institutes of Health (NIH) grants: F32GM140718 (from NIGMS to TMW), R35GM128562 (from NIGMS to BDF), F32GM136096 and K99/R00ES034058 (from NIGMS and NIEHS to KPMM), R01ES030575 (From NIEHS to DC) and R01ES032814 and R01CA269784 (from NIEHS and NCI to SAR). KPMM was additionally supported by the Office of the Vice Chancellor for Research and Graduate Education, University of Wisconsin-Madison and a Vanderbilt University Destination Biochemistry Advanced Postdoctoral Scholars Award. DC obtained funding from the Vanderbilt-Ingram Cancer Center. SAR received startup funds from the University of Vermont and the University of Vermont Cancer Center.

## Author contributions

CC, KPMM, TMW, BDF, DC, and SAR designed the project. KPMM and TMW developed experimental systems. CC developed computational pipelines. CC, KPMM, TMW, JAL, BDF, and SAR performed experiments and analyzed data. CC, KPMM, TMW, BDF, DC, and SAR interpreted results. CC and TMW created figures. CC, KPMM, TMW, and SAR wrote the manuscript. CC, KPMM, TMW, BDF, DC, and SAR edited the manuscript.

## Competing interests

The authors declare no competing interests.
