## [Transparent Peer Review file · Nature Communications]

Contributing factors to the oxidation-induced mutational landscape in human cells.

Corresponding Author: Dr Steven Roberts

Version 0:

Reviewer comments:

Reviewer #1

(Remarks to the Author)

This is a very well written and novel approach to determining the mechanistic components of oxidation-induced mutational spectra in human cells. The authors address a long-standing issue in the field of mutagenesis, which is how and through which DNA repair enzymes do reactive oxygen species cause particular mutations and patterns. They use a powerful approach by performing whole-genome sequencing after ROS treatment and bioinformatically determining mutational signatures and matching to, for example, SBS18. This extends work published previously by Stratton et al., but much further information is gathered by looking specific DNA repair mutants / knockouts, and performing biochemical and crystallography experiments. This is an important advance and contribution to the literature.

(Remarks on code availability)

Reviewer #2

(Remarks to the Author)

8-oxoG is a common single-base DNA lesion caused by oxidative stress, which can be recognized and excised by the glycosylase OGG1 to initiate the base excision repair pathway. In this manuscript, Cordero et al. characterized the KBrO₃ induced 8-oxoG DNA lesion and the resulting DNA mutations in the genome level and in the chromatin/nucleosome context, and explored the structural mechanism of OGG1 in recognizing and handling the 8-oxoG base embedded in the nucleosome. Overall, this manuscript provides novel information on the OGG mediated DNA repair pathway.

However, some of the conclusions including the title appeared to be overstated, and should be revised. The title "Hierarchical determinants of the oxidation-induced mutational landscape" is not fully supported by the assays in this manuscript and should not be the major conclusion. While the "oxidation-induced mutational landscape" was well studied in this manuscript, the observations from the assays on the three layer/Hierarchical determinants were over-interpreted and these conclusions were drawn more on the known possible roles of these related factors or processes. It's better for the manuscript to focus on the uncovered mutational landscape and the structural observations, and to shorten the discussion about the hierarchical determinants.

Specific concerns:

1. Page 9, the first paragraph and Figure 2, the conclusion on the similarity between "the difference of the KBrO₃ and endogenous 8-oxoG proportional sequence contexts" and "that of the KBrO₃-induced mutations and SBS18". (1) In Figure 2A, the trinucleotide preferences of endogenous lesions in Hela cells from a previous work were summarized. Do Hela cells also show SBS18 signature? If not, the similarity comparison was not appropriate. (2) The comparison of percentage differences in Figure 2B appeared to be problematic. Are there any data normalization procedures applied?

2. Page 9, the second paragraph, "profiled 8-oxoG lesions and mutations in WT RPE-1 cells..." Was 8-oxoG lesions also from WT RPE-1, or from Hela cell? Please clarify.

3. Page 11, the second paragraph and Figure 4, many key residues of OGG1 interacting with DNA were shown and described. Considering the 5-7 angstrom resolution of OGG1, could the side chain of these residues be accurately

assigned? Please show the local densities of these residues.

4. In Page 12, the first paragraph, what did the “1 bp rotation” mean? 1 bp register shift?

5. Figure 5A, S11A, why only one sample of POLH^{-/-} cell line was detected? Considering the large variations of the other cell lines, one POLH^{-/-} sample is not sufficient to draw conclusion.

6. Some of the conclusions drawn in the manuscript were not directly supported by the experimental observations, for instance:

Page 15, the first paragraph, “8-oxoG-induced mutations however displayed slightly more G>T substitutions on the leading strand suggesting that a DNA repair process (potentially MMR) preferentially removes 8-oxoG from the lagging strand template”;

“HMCES deletion exacerbated this effect, indicating that HMCES may favor bypass of leading strand lesions”;

“Interestingly, loss of Pol η removed the replication strand asymmetry. This indicates that Pol η likely mediates error-free bypass of 8-oxoG in the lagging strand template in human cells”.

Moreover, as shown in Figure 6, the differences of the mutation numbers between in leading strand and in lagging strand or between in non-transcribed and in transcribed were very small for WT and HMCES^{-/-} cell lines, were the differences statistically significant?

(Remarks on code availability)

Reviewer #3

(Remarks to the Author)

This manuscript investigates the mutational landscape induced by potassium bromate (KBrO₃) using immortalized RPE-1 cells. The authors established the chromatin structural determinants of 8-oxodG repair and elucidated three mechanisms involved in the repair of KBrO₃-induced 8-oxodG mutations.

The paper by Codero C. et al. demonstrated that the mutational pattern from KBrO₃ exposure differs slightly from the known endogenous ROS mutational signature, COSMIC SBS18. They identified a novel INDEL signature attributed to KBrO₃ exposure, characterized by C and T deletions, whose mechanism remains unknown. The authors profiled KBrO₃-induced lesions in knockout (KO) cells for OGG1, MUTYH, Pol η , and HMCES. They described the DNA repair processes of 8-oxodG through base excision repair (BER), considering factors such as chromatin state, nucleosome binding, transcription factor binding, replication asymmetry, and transcription strand bias.

From a broader perspective, these findings significantly enhance our understanding of the DNA damage and repair mechanisms induced by oxidants. Overall, the manuscript is well-written, the analyses are expertly performed, and the interpretations are well-founded.

Comments:

Major:

Codero C. et al., have found that KBrO₃-induced mutagenesis differs than endogenous ROS-induced mutagenesis (SBS18), in regards to sequence context. Is this applicable only for sequence context? What about replication asymmetry and TF binding? Using the generated data, it is possible to compare the behavior of no treatment (NT) samples with KBrO₃ exposed and explore replication asymmetry in the KO cell lines. Similarly, how does the spectra of INDELS look like in the NT after KO?

Pol ^{-/-} cells exhibited an increase of SBS and indels count. Nevertheless, when we look closely, we notice a probable decrease in C>A transversions, while there's an increase of C>G and C>T mutations, which could be attributed to clock-like signatures instead. I recommend plotting fold-change difference between the WT and Pol ^{-/-} cells for each mutation type, and performing a COSMIC deconvolution to elucidate the changes after KO.

Additionally, we note the absence of ID1 and ID2 (attributed to replication slippage) in the INDEL mutational pattern of Pol ^{-/-} cells, while it is still present in HMCES^{-/-} cells. If Figure 5C represents the INDEL mutational signatures, then show their activities, and whether the ID1 and ID3 are indeed not present in these cells. This could add to polymerases involved in replication slippage.

Mutations caused by endogenous ROS are likely to accumulate gradually over time and might be present at lower allelic frequency due to their random and continuous occurrence throughout the lifespan of the cell. By calculating the allelic frequency of C>A mutations we can understand the timing and clonality of the C>A mutations and the difference between SBS18 in NT (culture artifacts) and exogenous treatment to KBrO₃.

To elucidate the contribution of ROS to the mechanism of KBrO₃-induced mutagenesis, it would be great to measure ROS formation upon KBrO₃ treatment. Understanding why these two mutational processes, despite generating the same DNA lesion (8-oxoG), result in different mutational contexts warrants an explanation. A targeted experiment would involve using sensitive assays, such as electron paramagnetic resonance spectroscopy or fluorescent ROS indicators, to detect and

quantify the specific ROS generated during KBrO3 exposure. This approach can help pinpoint which ROS are driving the unique mutational signature associated with KBrO3 and exclude those that do not play a significant role.

A recent paper by Zhivagui et al. utilized UVA radiation to generate ROS in human cells and subsequent 8-oxodG formation in human cells. These cells exhibited SBS18 enrichment compared to control samples. This study should be cited to illustrate the association of exogenous factors with ROS production, 8-oxoG formation, and COSMIC SBS18 attribution. Additionally, an explanation for the use of RPE-1 cells in this context is needed.

Additionally, a reference is missing regarding the presence of SBS18 in control samples and its attribution to spontaneous culture artefacts. This was evidenced in Zhivagui et al., which found 23% of mutations attributed to SBS18 in control samples.

Minor:

Supplementary Figures 2 and 3 could be merged for better alignment with the text.

Figure 1E should name the INDEL signature "ID83D" to avoid confusion with COSMIC ID4.

In Figure 2, it is unclear why CLAPS-seq reads from HeLa cells were used when the authors later generated their own RPE-1 derived 8-oxoG lesions (and mutations) upon KBrO3 exposure.

For clarity, each plot in every figure should specify whether the data is generated from NT or KBrO3 treatment.

Confusing: HMCES deficiency led to increased mutagenicity which is in line with the fact that HMCES prevents mutagenesis. What is surprising here?

(Remarks on code availability)

Code looks good.

In the README file:

Cloning the repository using git:

```
`git clone https://github.com/yourusername/your-repo-name.git`
```

should be replaced with:

```
`git clone https://github.com/S-RobertsLab/Cordero-et-al.-2024.git`
```

Version 1:

Reviewer comments:

Reviewer #2

(Remarks to the Author)

I have no further questions. The authors have addressed my concerns.

(Remarks on code availability)

Reviewer #3

(Remarks to the Author)

The authors have done a good job improving this manuscript. They have increased the significance and the stats for this paper, and have also carefully addressed the more specific points. On my opinion, the study is comprehensive and interesting to warrant publication.

Minor:

Please be careful of typos, especially when writing "strand". In Page 19, it writes "strain bias" instead of "strand bias".

Thank you.

(Remarks on code availability)

Reviewer #1 (Remarks to the Author): expert in BER

This is a very well written and novel approach to determining the mechanistic components of oxidation-induced mutational spectra in human cells. The authors address a long-standing issue in the field of mutagenesis, which is how and through which DNA repair enzymes do reactive oxygen species cause particular mutations and patterns. They use a powerful approach by performing whole-genome sequencing after ROS treatment and bioinformatically determining mutational signatures and matching to, for example, SBS18. This extends work published previously by Stratton et al., but much further information is gathered by looking specific DNA repair mutants / knockouts, and performing biochemical and crystallography experiments. This is an important advance and contribution to the literature.

Response: We thank the reviewer for their positive feedback on our manuscript and their belief that this work represents an important contribution to the literature.

Reviewer #2 (Remarks to the Author): expert in structural biology of nucleosomes, cryo-EM

8-oxoG is a common single-base DNA lesion caused by oxidative stress, which can be recognized and excised by the glycosylase OGG1 to initiate the base excision repair pathway. In this manuscript, Cordero et al. characterized the KBrO₃ induced 8-oxoG DNA lesion and the resulting DNA mutations in the genome level and in the chromatin/nucleosome context, and explored the structural mechanism of OGG1 in recognizing and handling the 8-oxoG base embedded in the nucleosome. Overall, this manuscript provides novel information on the OGG mediated DNA repair pathway.

Response: We thank the reviewer for their appreciation of the novelty of our results.

However, some of the conclusions including the title appeared to be overstated, and should be revised. The title “Hierarchical determinants of the oxidation-induced mutational landscape” is not fully supported by the assays in this manuscript and should not be the major conclusion. While the “oxidation-induced mutational landscape” was well studied in this manuscript, the observations from the assays on the three layer/Hierarchical determinants were over-interpreted and these conclusions were drawn more on the known possible roles of these related factors or processes. It’s better for the manuscript to focus on the uncovered mutational landscape and the structural observations, and to shorten the discussion about the hierarchical determinants.

Response: We agree with the reviewer that our interpretations of a hierarchical organization of different oxidative DNA damage repair mechanisms is inferred based upon mutational biases uncovered in single knockout cell lines. While these data provide information on how individual repair mechanisms limit 8-oxoG mutagenesis, they do not necessarily indicate an ordered interaction between repair pathways. We have changed the title of the manuscript to: “Contributing factors to the oxidation-induced mutational landscape in human cells” and significantly shortened the discussion section on “An expanded 8-oxoG repair network in human cells” to more clearly state that we see evidence of three types of 8-oxoG repair/bypass, however, these may not work in a coordinated manner.

Specific concerns:

1. Page 9, the first paragraph and Figure 2, the conclusion on the similarity between “the difference of the KBrO₃ and endogenous 8-oxoG proportional sequence contexts” and “that of the KBrO₃-induced mutations and SBS18”. (1) In Figure 2A, the trinucleotide preferences of endogenous lesions in HeLa cells from a previous work were summarized. Do HeLa cells also show SBS18 signature? If not, the similarity comparison was not appropriate. (2) The comparison of percentage differences in Figure 2B appeared to be problematic. Are there any data normalization procedures applied?

Response: CLAPS-seq measures 8-oxoG lesions existing in a current population of cells, meaning that the endogenous 8-oxoG lesions measured in HeLa cells were present at the time the cells were cultured. The presence of these lesions in HeLa cells does not necessarily relate to past mutagenic lesion generation in the cell line or the tumor it was derived from. Instead, we used the sequence contexts of the endogenous 8-oxoG lesions to determine sequences where 8-oxoG lesions derived from endogenous reactive oxygen species are most likely to form and determined that they were different from where 8-oxoG lesions produced from KBrO₃-treatment form. This is consistent with KBrO₃ generating a 8-oxoG by a different mechanism than

traditional ROS¹, which we have emphasized in response to a comment by Reviewer 3. We edited the manuscript to clarify that the CLAPS-seq data is being used only to determine sequence preference of 8-oxoG derived from endogenous ROS. This section now reads: “We therefore obtained CLAPS-seq reads, which identify the genomic positions (at single nucleotide resolution) of 8-oxoG lesions formed during culture, from HeLa cells grown in the presence and absence of KBrO₃³⁹. Like mutations, KBrO₃-induced 8-oxoG occurred at a different distribution of sequence contexts compared to 8-oxoG caused by endogenous ROS in untreated HeLa cells (cosine similarity = 0.909) (Figure 2B).”

We still obtained somatic mutation calls for HeLa cells from the Cancer Cell Line Encyclopedia to assess whether HeLa cells have experienced 8-oxoG-induced mutations in the past. The spectra of these mutations did contain C to A substitutions, which could be caused by 8-oxoG, however, the limited number of mutations in this exome sequencing data limits our ability to definitively state this.

[REDACTED]

Above is the spectra of somatic mutations in HeLa cells generated from DepMap’s list of somatic mutations in HeLa cells.

We also recognized that the y-axes in Figure 2A were on different scales, which was visually confusing for comparing the percentage of lesions and mutations in different sequence contexts in Figure 2B. We have fixed all Figure 2A axes to a maximum of 30%, which enables easy comparison of bar heights and simplifies the conversion of this data into the differences shown in Figure 2B. No data normalization procedures were applied in generating the differences of the percent spectra graphs in Figure 2B. These bar heights result directly from the subtraction of the respective graphs in Figure 2A.

2. Page 9, the second paragraph, “profiled 8-oxoG lesions and mutations in WT RPE-1 cells...” Was 8-oxoG lesions also from WT RPE-1, or from HeLa cell? Please clarify.

Response: We clarified this statement to indicate that the 8-oxoG lesions were measured in KBrO₃ treated HeLa cells, while mutation data was from KBrO₃-treated RPE-1 cells. The text now reads: “To accomplish this, we profiled where KBrO₃ treatment forms 8-oxoG lesions and mutations (using HeLa cell CLAPS-seq data and WT RPE-1 cell variant calls, respectively) relative to different chromatin states derived by Hidden Markov Modeling (HMM) of eight histone modifications and CTCF.”

3. Page 11, the second paragraph and Figure 4, many key residues of OGG1 interacting with DNA were shown and described. Considering the 5-7 angstrom resolution of OGG1, could the side chain of these residues be accurately assigned? Please show the local densities of these residues.

Response: As noted by the reviewer, the local resolution of OGG1 (5 - 7 Å) in the cryo-EM maps is not sufficient for accurately assigning the exact positioning of OGG1 side chains. The OGG1 side chain conformations within both structures in the manuscript reflect those from the

high-resolution structure of OGG1 bound to non-nucleosomal DNA (PDB: 1EBM). We rationalized this decision as key OGG1 secondary structural elements from the high-resolution structure of OGG1 bound to non-nucleosomal DNA (PDB: 1EBM), where side chain conformation could be observed, fit extremely well into the cryo-EM map. This suggests that nucleosome binding by OGG1 does not significantly alter the OGG1 structure seen in the high-resolution crystal structure (PDB: 1EBM). Consistently, the all-atom clash score for the OGG1-8oxoG-NCP-6 and OGG1-8oxoG-NCP+4 structures are 7 and 6, respectively, which we reasoned would be significantly higher if the side chain conformations near the nucleosomal DNA and 8oxoG base were not plausible.

To rationalize this decision and to provide the necessary context for future readers of our manuscript, we have added the following statement in the main text for OGG1-8oxoG-NCP-6 structure; *“Although the local resolution of OGG1 (5 - 7 Å) is not sufficient for determining the exact position of OGG1 side chains, the side chain conformations presented below represent those from the high-resolution X-ray crystal structure of OGG1 (PDB: 1EBM).”* We have also added the following statement in the main text for OGG1-8oxoG-NCP+4 structure; *“Importantly, the cryo-EM map was sufficient to dock the previously determined high-resolution X-ray crystal structure of OGG1 (PDB: 1EBM)² into the cryo-EM map (Supplemental Figure 9H), and the side chain conformations were kept from the high-resolution X-ray crystal structure of OGG1 (PDB: 1EBM).”*

We also apologize to the reviewer that the cryo-EM maps, models, and deposition reports associated with the structures were not uploaded with the initial submission, as the file sizes were too large for directly uploading to *Nature Communications*. We have now provided the deposition reports with the manuscript resubmission and have released all the cryo-EM maps and models from the EMDB and PDB, respectively (date of release: 10/17/2024). This will allow the reviewers to assess the cryo-EM data quality and the conclusions drawn from the cryo-EM data throughout the manuscript.

4. In Page 12, the first paragraph, what did the “1 bp rotation” mean? 1 bp register shift?

Response: We apologize for the lack of clarity in this description for the conformational changes in the nucleosomal DNA observed upon OGG1 binding. The “1-bp rotation” is indeed a 1-bp register shift in the nucleosomal DNA. We have replaced “1-bp rotation” in the manuscript text with “1-bp register shift in the nucleosomal DNA.” We have also updated the nomenclature used in Fig. 4D.

5. Figure 5A, S11A, why only one sample of POLH-/- cell line was detected? Considering the large variations of the other cell lines, one POLH-/- sample is not sufficient to draw conclusion.

Response: Figure 5A, S11A only contains one datapoint for the POLH-/- cell line because the data is an analysis of previously published sequencing from a POLH-/- cell treated with KBrO₃ (Yurchenko, A.A., Rajabi, F., Braz-Petta, T. et al. Genomic mutation landscape of skin cancers from DNA repair-deficient xeroderma pigmentosum patients. *Nat Commun* 14, 2561 (2023). <https://doi.org/10.1038/s41467-023-38311-0>). This publication used POLH-/- in RPE-1 cells and a similar KBrO₃ treatment protocol to our WT and HMCES-/- RPE-1 cells, making the mutation calls highly comparable. However, the publication only sequenced one outgrowth clone, which limited our analysis. The lack of additional samples means we cannot state with statistical certainty that the total number of mutations in POLH-/- cells are elevated compared to WT KBrO₃ cells. We have updated the text to clarify this point. While we cannot statistically conclude that POLH-/- elevates overall KBrO₃-induced mutagenesis from the number of

mutations observed, we added statistical comparisons of the changes in mutation distribution within the POLH^{-/-} sample. The proportions of specific substitution types (i.e. C to A, C to G, and C to T) in the POLH^{-/-} cells is statistically different compared to WT cells using a 2-way ANOVA (p=0.0345) (new Figure Supplemental Figure 14). Similarly, the changes in replication and transcription strand biases are also now supported by statistics, which allows us to conclude that loss of POLH changes the distribution of KBrO₃ mutations. Such changes are most likely to be mediated through an increase in mutation rate. We believe this is likely the case in the POLH^{-/-} cells.

6. Some of the conclusions drawn in the manuscript were not directly supported by the experimental observations, for instance:

Page 15, the first paragraph, “8-oxoG-induced mutations however displayed slightly more G>T substitutions on the leading strand suggesting that a DNA repair process (potentially MMR) preferentially removes 8-oxoG from the lagging strand template”;

“HMCEs deletion exacerbated this effect, indicating that HMCEs may favor bypass of leading strand lesions”;

“Interestingly, loss of Pol η removed the replication strand asymmetry. This indicates that Pol η likely mediates error-free bypass of 8-oxoG in the lagging strand template in human cells”.

Moreover, as shown in Figure 6, the differences of the mutation numbers between in leading strand and in lagging strand or between in non-transcribed and in transcribed were very small for WT and HMCEs^{-/-} cell lines, were the differences statistically significant?

Response: We agree that the differences in replication strand bias are very small in the WT, HMCEs^{-/-}, and POLH^{-/-} cells. We now include statistics to provide support that these differences are significant. We also included replication strand asymmetries for APOBEC cytidine deaminase-induced mutations (which are known to occur primarily on the lagging strand template) to provide a comparison of effect size (see Supplemental Figure 17A). We have added text at multiple places in the indicated section of text to emphasize that we observe statistically significant patterns in the mutation distributions that are different between the KBrO₃-treated WT, HMCEs^{-/-}, and POLH^{-/-} cells, but these preferences for mutations on the leading or lagging strands is only consistent with a model of differential lesion bypass for HMCEs on the leading strand and POLH on the lagging strand.

Reviewer #3 (Remarks to the Author): expert in mutational signatures and DNA damage, ROS and 8-oxoG DNA repair

This manuscript investigates the mutational landscape induced by potassium bromate (KBrO₃) using immortalized RPE-1 cells. The authors established the chromatin structural determinants of 8-oxodG repair and elucidated three mechanisms involved in the repair of KBrO₃-induced 8-oxodG mutations.

The paper by Codero C. et al. demonstrated that the mutational pattern from KBrO₃ exposure differs slightly from the known endogenous ROS mutational signature, COSMIC SBS18. They identified a novel INDEL signature attributed to KBrO₃ exposure, characterized by C and T deletions, whose mechanism remains unknown. The authors profiled KBrO₃-induced lesions in knockout (KO) cells for OGG1, MUTYH, Pol η, and HMCES. They described the DNA repair processes of 8-oxodG through base excision repair (BER), considering factors such as chromatin state, nucleosome binding, transcription factor binding, replication asymmetry, and transcription strand bias.

From a broader perspective, these findings significantly enhance our understanding of the DNA damage and repair mechanisms induced by oxidants. Overall, the manuscript is well-written, the analyses are expertly performed, and the interpretations are well-founded.

Response: We thank the reviewer for their complementary assessment of the quality and significance of the research.

Comments:

Major:

Codero C. et al., have found that KBrO₃-induced mutagenesis differs than endogenous ROS-induced mutagenesis (SBS18), in regards to sequence context. Is this applicable only for sequence context? What about replication asymmetry and TF binding? Using the generated data, it is possible to compare the behavior of no treatment (NT) samples with KBrO₃ exposed and explore replication asymmetry in the KO cell lines. Similarly, how does the spectra of INDELS look like in the NT after KO?

Response: SBS18 mutations have been previously reported to have minor replication and transcription asymmetry but are enriched in late replicating genome regions that generally correlate with heterochromatin regions^{3,4}. All three of these characteristics are therefore conserved between our KBrO₃-induced mutations and SBS18, indicating that they result mostly from repair processes operating on the 8-oxoG post formation. In contrast, the difference in sequence context for 8-oxoG formation induced by KBrO₃ and endogenous ROS likely results from differences in chemistry. While transcription factor binding could influence the local reactivity of a guanine in the binding site, we did not see any impact of TF binding on KBrO₃-induced mutagenesis. In line with this finding, the density of SBS18 mutations are unchanged in CTCF binding sites³. We have expanded the discussion of these similarities in the text by adding the following statement to the discussion: "In contrast to the variable sequence preferences for 8-oxoG formation induced by KBrO₃ or ROS, chromatin impacts on 8-oxoG induced mutation appear to be largely conserved for each method of lesion formation as the observed distributions of mutation with respect to chromatin state, nucleosome occupancy, replication and transcriptional strain bias, and transcription factor binding sites for KBrO₃-induced mutation largely mirror those reported for SBS18 mutations in human tumor genomes^{10,30}."

We also analyzed replication and transcriptional asymmetries in NT KO cell lines. Overall, the number of mutations in these analyses were 10-fold lower, which limits what can be confidently discerned. We did have sufficient G>T substitutions in the OGG1^{-/-}, MUTYH^{-/-}, HMCES, and PolH^{-/-} lines to determine that all display a slight leading strand bias similar to WT cells. Interestingly, the OGG1^{-/-} KO NT cells also displayed transcriptional asymmetry favoring the non-transcribed strand. This pattern would be consistent with TC-NER functioning in the absence of OGG1-mediated BER to preferentially repair the transcribed strand of genes. Effect is similar to what we observed with KBrO₃ treated POLH^{-/-} cells. The NT mutational asymmetry data has been added as Supplemental Figures 17 and 18.

We included the spectra of INDELs in NT KO cells in Supplemental Figure 13. All KO cells displayed INDEL spectra primarily composed of T insertion and T deletions in homopolymer runs.

Pol h^{-/-} cells exhibited an increase of SBS and indels count. Nevertheless, when we look closely, we notice a probable decrease in C>A transversions, while there's an increase of C>G and C>T mutations, which could be attributed to clock-like signatures instead. I recommend plotting fold-change difference between the WT and Pol h^{-/-} cells for each mutation type, and performing a COSMIC deconvolution to elucidate the changes after KO.

Response: Instead of plotting fold-change between WT and Pol h^{-/-} cells, we graphed the total number per genome of each mutation type for WT and Pol h^{-/-} cells. This showed that C>A, C>G, and C>T mutations all increased in the Pol h^{-/-} cells compared to WT cells. We have added this data to Supplemental Figure 14.

Additionally, we note the absence of ID1 and ID2 (attributed to replication slippage) in the INDEL mutational pattern of Pol h^{-/-} cells, while it is still present in HMCES^{-/-} cells. If Figure 5C represents the INDEL mutational signatures, then show their activities, and whether the ID1 and ID3 are indeed not present in these cells. This could add to polymerases involved in replication slippage.

Response: Because INDEL mutations calls are more susceptible to differences in sequencing coverage and the software used for variant analysis, we suspected the lack of ID1 and ID2-like mutations in the Pol h^{-/-} cells may be due to our original utilization of published mutation calls for this data, which used a different mutation calling methodology. We therefore recalled mutations for the KBrO₃ Pol H^{-/-} cells using our variant calling pipeline. This produced very similar data for substitution mutations, maintained the previously observed C and T deletions, but added additional ID1 and ID2 like mutations. We therefore believe the original lack of these mutations was due to differences in variant calling methods and not due to a role of pol eta in replication slippage. We have added the recalled variant numbers and spectra in Figure 5.

Mutations caused by endogenous ROS are likely to accumulate gradually over time and might be present at lower allelic frequency due to their random and continuous occurrence throughout the lifespan of the cell. By calculating the allelic frequency of C>A mutations we can understand the timing and clonality of the C>A mutations and the difference between SBS18 in NT (culture artifacts) and exogenous treatment to KBrO₃.

Response: We agree that this would be an interesting analysis, however, analysis of allelic fraction to determine mutational timing is not possible in the cell line based mutational experiments we performed. This is because we had to obtain clonal isolates following the mutation accumulation phase of our experimental design (shown in Figure 1) to identify mutations by Illumina whole genome sequencing, The clonal isolation of cells following KBrO₃

treatment results all mutations accumulated in throughout the treatment to be fixed at a 50% allelic fraction.

To elucidate the contribution of ROS to the mechanism of KBrO₃-induced mutagenesis, it would be great to measure ROS formation upon KBrO₃ treatment. Understanding why these two mutational processes, despite generating the same DNA lesion (8-oxoG), result in different mutational contexts warrants an explanation. A targeted experiment would involve using sensitive assays, such as electron paramagnetic resonance spectroscopy or fluorescent ROS indicators, to detect and quantify the specific ROS generated during KBrO₃ exposure. This approach can help pinpoint which ROS are driving the unique mutational signature associated with KBrO₃ and exclude those that do not play a significant role.

Response: We have now completed measurement of cellular ROS levels induced by KBrO₃ using CellROX 488 staining. This analysis indicated that KBrO₃ treatment increases cellular ROS levels, however, the effect is modest (new Supplemental Figure 1). Such a result is consistent with previously published data indicating that KBrO₃-induces 8-oxoG by a different chemistry than reactive oxygen species¹. KBrO₃-induced 8-oxoG requires glutathione and is not inhibited by traditional ROS scavengers like catalase or superoxide dismutase. We have added a discussion of these results to the text. The text now reads:

“KBrO₃ treatment resulted in only a modest increase in cellular ROS (~2-fold; new Supplemental Figure 1), This result is consistent with previous experiments showing that KBrO₃ induces 8-oxoG by a chemical reaction that requires glutathione and is resistant to catalase and superoxide dismutase ROS scavengers, therefore occurring distinctly from traditional ROS.”

A recent paper by Zhivagui et al. utilized UVA radiation to generate ROS in human cells and subsequent 8-oxodG formation in human cells. These cells exhibited SBS18 enrichment compared to control samples. This study should be cited to illustrate the association of exogenous factors with ROS production, 8-oxoG formation, and COSMIC SBS18 attribution. Additionally, an explanation for the use of RPE-1 cells in this context is needed.

Response: We have added a reference to the following sentence in the introduction. “ROS are generated in cells by endogenous processes like lipid peroxidation and cell metabolism^{2,3} or through exposure to exogenous chemicals agents such as potassium bromate (KBrO₃)^{1,4}, a former food additive, or UVA exposure⁵.”

We also added the following sentence: “This hypothesis is supported by experimental evidence indicating that deficiency in 8-oxoG repair mechanisms⁸ or UVA exposure⁵, which generates cellular ROS, resulting in SBS18-like mutations.”

We justified the use of RPE-1 cells with the following sentence added to the results: “RPE-1 cells were chosen for their diploid genome status, which facilitates mutation calling, and non-cancerous origin making these cells a closer model to normal cells in the body.”

Additionally, a reference is missing regarding the presence of SBS18 in control samples and its attribution to spontaneous culture artefacts. This was evidenced in Zhivagui et al., which found 23% of mutations attributed to SBS18 in control samples.

Response: We have added the requested reference at the end of the statement: “Deconvolution of these signatures into known COSMIC SBS signatures revealed that untreated RPE-1 cells contained a broad spectrum of base substitution mutations most consistent with SBS40, SBS5, and a small percentage of SBS18 (Supplemental Figure 2C), which are all consistent with spontaneously accumulated mutations during cell culture³⁴”

Minor:

Supplementary Figures 2 and 3 could be merged for better alignment with the text.

Response: We have added additional panels comparing our KBrO₃ INDEL signature to COSMIC ID5 to Supplemental Figure 2. The similarity of these two signatures and the correlation of ID5 with SBS18 in tumor cells strongly suggests that COSMIC ID5 is an oxidative DNA damage signature. We therefore have left Supplemental Figure 3 as a separate figure.

Figure 1E should name the INDEL signature "ID83D" to avoid confusion with COSMIC ID4.

Response: We have edited Figure 1E as requested.

In Figure 2, it is unclear why CLAPS-seq reads from HeLa cells were used when the authors later generated their own RPE-1 derived 8-oxoG lesions (and mutations) upon KBrO₃ exposure.

Response: We thank the reviewer for pointing out a confusingly written sentence in the manuscript. All 8-oxoG lesions analyzed in the manuscript were those identified with CLAPS-seq reads from HeLa cells. We did not generate our own RPE-1 derived 8-oxoG lesions. The sentence in question was also indicated as being confusing by reviewer 2. We have edited to the following to remove any ambiguity as to the source of the 8-oxoG lesion data. "To accomplish this, we profiled where KBrO₃ treatment forms 8-oxoG lesions and mutations (using HeLa cell CLAPS-seq data and WT RPE-1 cell variant calls, respectively) relative to different chromatin states derived by Hidden Markov Modeling (HMM) of eight histone modifications and CTCF."

For clarity, each plot in every figure should specify whether the data is generated from NT or KBrO₃ treatment.

Response: We have edited our figures as requested.

Confusing: HMCES deficiency led to increased mutagenicity which is in line with the fact that HMCES prevents mutagenesis. What is surprising here?

Response: We have edited the indicated sentence to the following for clarity. "HMCES-deficiency also resulted in elevated KBrO₃-induced mutagenesis indicating it also protects against 8-oxoG mutagenesis, consistent with the previously reported sensitivity of HMCES-/- cells to KBrO₃."

Reviewer #3 (Remarks on code availability):

Code looks good.

References:

1. Kawanishi, S., and Murata, M. (2006). Mechanism of DNA damage induced by bromate differs from general types of oxidative stress. *Toxicology* 221, 172-178. 10.1016/j.tox.2006.01.002.
2. Bruner, S.D., Norman, D.P., and Verdine, G.L. (2000). Structural basis for recognition and repair of the endogenous mutagen 8-oxoguanine in DNA. *Nature* 403, 859-866. 10.1038/35002510.

3. Otlu, B., Diaz-Gay, M., Vermes, I., Bergstrom, E.N., Zhivagui, M., Barnes, M., and Alexandrov, L.B. (2023). Topography of mutational signatures in human cancer. *Cell Rep* 42, 112930. [10.1016/j.celrep.2023.112930](https://doi.org/10.1016/j.celrep.2023.112930).
4. Pich, O., Muinos, F., Sabarinathan, R., Reyes-Salazar, I., Gonzalez-Perez, A., and Lopez-Bigas, N. (2018). Somatic and Germline Mutation Periodicity Follow the Orientation of the DNA Minor Groove around Nucleosomes. *Cell* 175, 1074-1087 e1018. [10.1016/j.cell.2018.10.004](https://doi.org/10.1016/j.cell.2018.10.004).

REVIEWERS' COMMENTS

Reviewer #2 (Remarks to the Author):

I have no further questions. The authors have addressed my concerns.

Response: We are happy to have addressed the reviewer's concerns.

Reviewer #3 (Remarks to the Author):

The authors have done a good job improving this manuscript. They have increased the significance and the stats for this paper, and have also carefully addressed the more specific points. On my opinion, the study is comprehensive and interesting to warrant publication.

Minor:

Please be careful of typos, especially when writing "strand". in Page 19, it writes "strain bias" instead of "strand bias".

Response: We thank the reviewer for their positive assessment of the revision and study overall. We have corrected the indicated typo and reviewed the entire document to correct any others that were identified.